# SceneLCM: Multi-Room Indoor Scene Generation with Latent Consistency Modeling

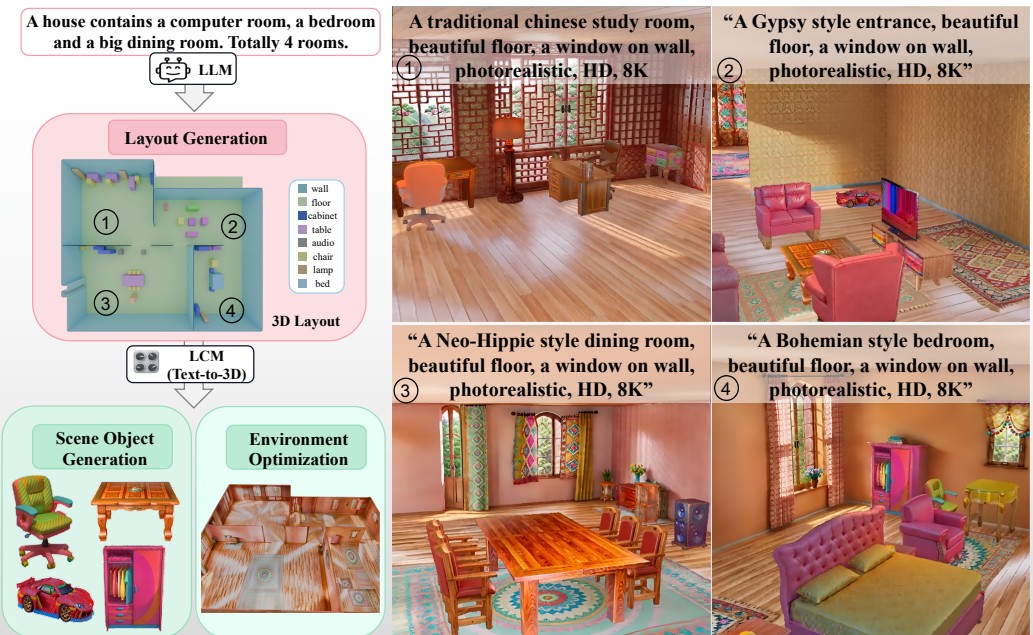

**Figure 1:** From a textual scene description, **SceneLCM** delivers fast, high-fidelity multi-room generation through an automatic and interactive pipeline uniting LLM-guided layout generation with latent consistency optimization of objects and environments.

## Abstract

Automatically generating complex, realistic, and interactive indoor scenes from user prompts remains a formidable challenge, requiring scalability to multi-room environments, physical plausibility, controllable editing, and minimal human intervention. Existing paradigms, such as text-to-3D synthesis and layout-based retrieval, provide complementary advantages but suffer from limited automation, structural incompleteness, suboptimal textures, and inefficiency in large-scale settings. To overcome these limitations, we introduce **SceneLCM**, an automatic and interactive framework that integrates Large Language Model (LLM)-driven layout generation with *Latent Consistency Model* (LCM)-based scene optimization. Central to SceneLCM is the proposed **Consistency Trajectory Sampling** (CTS) loss, which maintains self-consistency during LCM optimization, enabling faster convergence and higher-fidelity 3D generation with theoretically bounded distillation error. Built upon CTS-guided LCMs, the SceneLCM pipeline comprises three key stages: (1) **Layout Generation** — LLM-guided 3D spatial reasoning transforms textual descriptions into parametric floorplans and object configurations, refined via iterative programmatic verification and cluster-based object orientation; (2) **Scene Object Generation** — objects are represented as 3D Gaussians and optimized with CTS for efficient, photorealistic results; (3) **Environment Optimization** — a normal-aware texture field encodes multi-resolution scene appearance, optimized with CTS along Zigzag camera trajectories to ensure geometric and texture coherence. Extensive experiments demonstrate that SceneLCM

produces high-quality, diverse, and physically coherent single- and multi-room scenes, while supporting texture editing and physically plausible modifications. Ablation studies validate the critical role of CTS in enabling high-quality, rapid generation across all scene components. The implementation will be publicly released to support reproducibility and foster further research.

# 1 INTRODUCTION

Generating diverse indoor scenes is critical for Embodied AI and AR/VR applications, yet creating multi-room environments with physical realism remains highly challenging due to substantial knowledge requirements and computational costs (Community, 2018; Unity Technologies, 2023; Epic Games). Recent works have explored this problem through two main paradigms: *text-to-3D generation* (Yang et al., 2024a; Liang et al., 2024; Zhong et al., 2024; Li et al., 2024a;b) and *layout-based object retrieval* (Yang et al., 2024b; Fu et al., 2024; Wang et al., 2024b; Lin & Mu, 2024).

Despite notable progress, both paradigms face fundamental limitations. Text-to-3D methods achieve high-fidelity asset generation with photorealistic textures but struggle with generating large-scale multi-room environments, decoupling furniture components for parametric editing and physical plausibility, keeping optimization time manageable in large environments, and achieving full pipeline automation without requiring manual specification of layouts and camera trajectories (Yang et al., 2024a; Li et al., 2024a). In contrast, layout-based retrieval methods scale better for large scenes, but one-shot LLM prompting often leads to overlapping or misoriented objects, they frequently neglect texture optimization and omit essential structural elements such as windows, ceilings, and walls, and they rely heavily on large, annotated databases for inference.

To address these challenges, we propose an automatic, interactive indoor Scene generation pipeline using Latent Consistency Models, termed **SceneLCM**, which combines text-to-3D and layout-based generation with LCM (Song et al., 2023; Luo et al., 2023) to efficiently produce high-fidelity, realistic, and diverse multi-room scenes, as illustrated in Fig. 1.

SceneLCM leverages LCMs to enable fast and stable generation of 3D models, including scene objects and environments. As distillation sampling strongly influences generation quality and speed (Wu et al., 2024), we introduce a novel **Consistency Trajectory Sampling** (CTS) loss to guide LCM optimization. CTS preserves the self-consistency of LCMs for high-fidelity generation while enabling faster optimization. We further provide two theoretical justifications demonstrating the self-consistency and bounded distillation error of CTS.

Built upon CTS-guided LCMs, the automatic generation pipeline of SceneLCM consists of the following three main stages:

- **Layout Generation**: Given a user-provided scene description, LLMs are prompted to produce an initial layout, comprising the floorplan and object configurations, which is further refined through iterative programmatic verification and cluster-based orientation assignment (c.f. Fig. 2).

- **Scene Object Generation**: Scene objects are represented as 3D Gaussians and optimized with an LCM under the proposed CTS loss to generate photorealistic objects efficiently according to the specified texture descriptions (c.f. Fig. 3).

- **Environment Optimization**: A normal-aware texture field module models high-quality environment textures, which are optimized with a CTS-guided LCM by rendering scene images along a Zigzag camera trajectory to ensure comprehensive scene coverage (c.f. Fig. 4).

Extensive experiments are conducted to evaluate the effectiveness of SceneLCM in generating high-fidelity, realistic, and diverse scenes efficiently for both single-room and multi-room settings. SceneLCM also demonstrates capabilities for texture and physically plausible editing, highlighting its controllability and interactivity. Ablation studies further verify the effectiveness of individual components, particularly the proposed CTS, which is central to applying LCMs for both scene object generation and environment optimization.

## 2 RELATED WORK

**Text-to-3D Generation** Text-to-3D generation primarily builds on Score Distillation Sampling (SDS) (Poole et al., 2022; Wang et al., 2023a), which distills 2D text-to-image diffusion priors into differentiable 3D representations through rendering-based optimization. Follow-up works (Huang et al.; Zhu et al.; Lin et al., 2023; Wang et al., 2023b; Chen et al., 2023; Liang et al., 2024; Dai et al., 2025) improve fidelity and efficiency with various 3D representations, while recent studies (Wu et al., 2024; Li et al., 2024b; Zhong et al., 2024; Chen et al., 2024b) incorporate consistency-model ideas (Song et al., 2023) to enhance multi-view coherence, albeit mostly heuristically. In contrast, we formally connect our CTS loss to the consistency objective, providing a theoretical grounding that strengthens both structural and text–view alignment in 3D generation.

**Indoor Scene Generation** Indoor scene generation methods can be broadly divided into layout-based object retrieval and text-to-3D generation. Layout-based methods first generate a scene layout and then retrieve objects from a database. Early approaches (Lin & Mu, 2024; Zhai et al., 2024) used generative models (Goodfellow et al., 2014; Ho et al., 2020) for layout generation, but limited 3D scene datasets led to low robustness and novelty. More recent works (Fu et al., 2024; Yang et al., 2024b; Deng et al., 2025) leverage LLMs/VLMs (Brown et al., 2020; Achiam et al., 2023) and CLIP for object retrieval from text prompts, producing editable scenes but often neglecting texture optimization, requiring annotated databases, and omitting key elements such as windows or walls.

Text-to-3D approaches (Zhang et al., 2024; Höllein et al., 2023; Schult et al., 2024) can generate realistic visuals via image inpainting but suffer from limited 3D consistency and lack furniture editability. Some methods (Li et al., 2024a; Yang et al., 2024a) integrate layout guidance and SDS loss for more controllable and realistic generation but cannot autonomously generate layouts, handle many objects, or support editing. Our method addresses these limitations by providing an automatic framework for controllable, high-detail scene generation with physically plausible editing.

## 3 CONSISTENCY TRAJECTORY SAMPLING FOR CONSISTENCY MODELS

### 3.1 PRELIMINARY

**Consistency Model.** Consistency Models (CMs) (Song et al., 2023) is proposed to enable single-step or few-step generation by distilling knowledge from a pre-trained diffusion model (DM) (Ho et al., 2020). The core idea is to learn a consistent function $f_\theta(\cdot, \cdot)$ with trainable parameters $\theta$, which directly predicts the denoised image $x_0$ given a noisy sample $x_t$ and its corresponding time step $t$. The model is trained by minimizing the self-consistency distillation loss:

$$L_{CD}(\theta, \theta^-) = \mathbb{E}\Big[w(t)\,\|f_\theta(x_{t_{n+1}}, t_{n+1}) - f_{\theta^-}(\hat{x}_{t_{n+1} \to t_n}, t_n)\|_2^2\Big], \tag{1}$$

where $0 = t_1 < t_2 < \cdots < t_N = T$, and $\hat{x}_{t_{n+1} \to t_n}$ is computed using the ODE solver $\Phi(\cdot)$. Here, $\theta^-$ denotes the parameters of target model, updated via an exponential moving average (EMA) of $\theta$. The objective of CM is to enforce the self-consistency condition along the probability flow ODE trajectory $\{x_t\}_{t \in [0,T]}$, i.e., $f(x_t, t) = f(x_{t'}, t') \quad \forall t, t' \in [0, T]$. In practice, the consistent function $f$ is parameterized through the noise prediction network $\epsilon_\theta$ as:

$$f(x_t, t) = c_{\text{skip}}(t)\, x_t + c_{\text{out}}(t)\, \frac{x_t - \sigma_t\, \epsilon_\theta(x_t, t)}{\alpha_t}, \tag{2}$$

where $c_{\text{skip}}(t)$ and $c_{\text{out}}(t)$ are differentiable functions, and $\alpha_t, \sigma_t$ define the noise schedule.

**Latent Consistency Model.** While CMs operate directly in the pixel space, Latent Consistency Models (LCMs) (Luo et al., 2023) extend this framework to the latent space of a pre-trained autoencoder. Specifically, a diffusion process is defined on the latent variable $z_t$, which is computationally more efficient and semantically richer than pixel representations. LCMs inherit the self-consistency principle of Eq. (1), but replace $x_t$ with $z_t$, leading to the latent consistency distillation loss:

$$L_{LCD}(\theta, \theta^-) = \mathbb{E}\Big[w(t)\,\|f_\theta(z_{t_{n+1}}, t_{n+1}) - f_{\theta^-}(\hat{z}_{t_{n+1} \to t_n}, t_n)\|_2^2\Big], \tag{3}$$

where $\hat{z}_{t_{n+1} \to t_n}$ is obtained via the ODE solver in the latent space. The model thus learns a consistent mapping $f_\theta(z_t, t)$ that directly predicts the clean latent $z_0$.

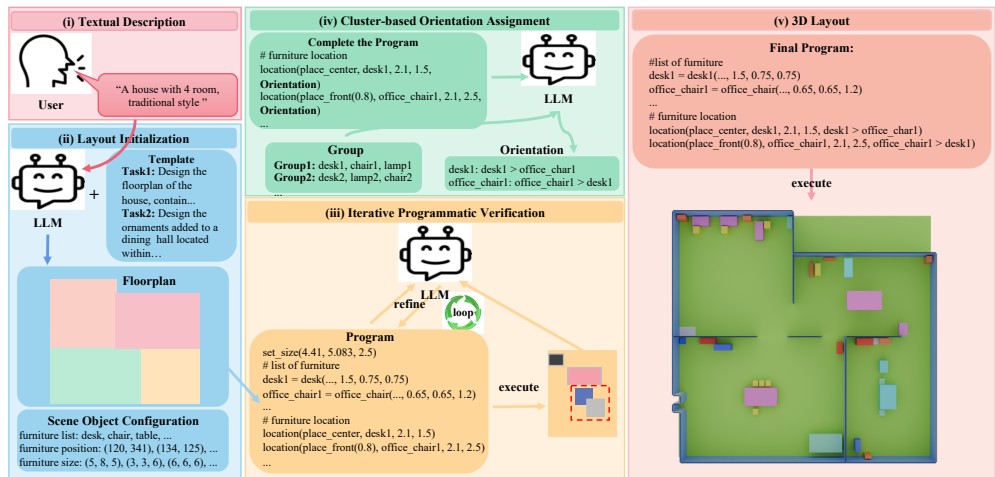

**Figure 2: An illustration of Layout Generation.** Given (i) a free-form textual input, we (ii) prompt LLMs to initialize the scene layout, including the floorplan and object configurations, (iii) apply iterative programmatic verification until a conflict-free layout is obtained, and (iv) assign object orientations via cluster-based reasoning, ultimately (v) producing a valid multi-room 3D layout.

## 3.2 CONSISTENCY TRAJECTORY SAMPLING

In this work, we adopt CMs to optimize 3D models due to their ability to enable fast and stable sampling. However, directly employing the original self-consistency distillation loss $L_{CD}$ in Eq. (1) presents two limitations. First, when computing gradients with $L_{CD}$, the Jacobian term of the entire function $f$ is typically omitted, which introduces approximation errors. In fact, several components of the Jacobian (e.g., $c_{skip}(t)$, $c_{out}(t)$, $\alpha_t$ and $\sigma_t$ in Eq. (2)) can be computed exactly and therefore should not be neglected. Second, $L_{CD}$ is incompatible with auxiliary techniques such as perpendicular negative sampling (Perp-Neg), which require separating and explicitly handling the noise term predicted by the noise prediction model $\epsilon_\theta$.

To address these issues, we derive a novel **Consistency Trajectory Sampling (CTS)** loss from $L_{CD}$ that preserves the self-consistency property of CMs while decoupling the noise term:

$$L_{CTS} = \mathbb{E}\Big[\underbrace{\|w_1(t)\big(\epsilon_\theta(x_{s\to t}, t, y) - \epsilon_\theta(x_s, s, \emptyset)\big)\|_2^2}_{\text{term1}} + \underbrace{\|w_2(s, t)\big(\epsilon_\theta(x_s, s, \emptyset) - \epsilon\big)\|_2^2}_{\text{term2}}\Big],$$

$$\text{with} \quad w_1(t) = c_{out}(t)\Big(\frac{\sigma_t}{\alpha_t}\Big), \quad w_2(s, t) = \big[c_{out}(t) - c_{out}(s)\big]\Big(\frac{\sigma_s}{\alpha_s}\Big). \tag{4}$$

Here, $t > s$ are adjacent diffusion steps, $x_s = \alpha_s x_\pi + \sigma_s \epsilon$, and $x_{s\to t}$ denotes a less noisy sample obtained via one deterministic ODE discretization step from $x_s$. The random noise $\epsilon$ is sampled once and fixed throughout optimization, while $y$ denotes the text condition. Intuitively, the two terms in Eq. (4) play complementary roles: term1 enforces local self-consistency within a sub-trajectory, while term2 promotes global cross-step consistency along the entire ODE trajectory. For LCMs, the CTS loss can be naturally extended by replacing $x_t$ with $z_t$, analogous to Eq. (3).

**Justification.** We provide two theoretical justifications: (i) the proposed CTS loss is mathematically equivalent to the Consistency Loss (Song et al., 2023), and (ii) upon convergence, it ensures the generation of high-fidelity 3D models. These results correspond to Theorem 1 and Theorem 2 in the Appendix, where detailed proofs are provided.

## SCENELCM: SCENE GENERATION WITH LATENT CONSISTENCY MODEL

As shown in Fig. 1, SceneLCM generates indoor scenes through three stages: Layout Generation, Scene Object Generation, and Environment Optimization. First, a scene description is used to prompt LLMs to produce a detailed layout, refined by iterative programmatic verification and cluster-based orientation assignment. Second, high-fidelity 3D scene objects are generated using 3D

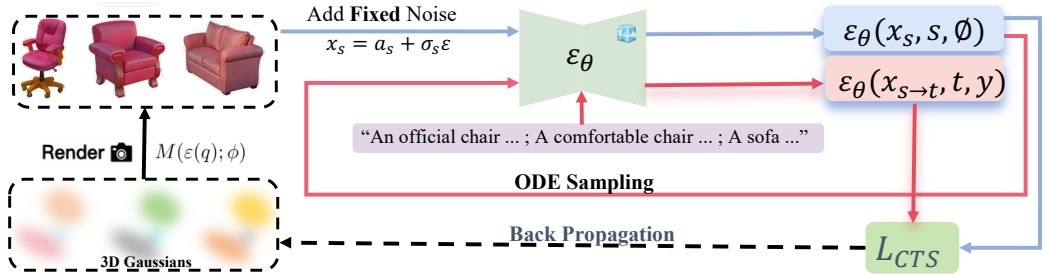

**Figure 3: An illustration of Scene Object Generation.** Scene objects are represented as 3D Gaussians and optimized with Latent Consistency Model under our proposed Consistency Trajectory Sampling loss.

Gaussians optimized with Latent Consistency Models (LCMs) and the proposed Consistency Trajectory Sampling (CTS) loss. Third, the environment is textured and optimized using a Normal-Aware Texture Field Module under CTS-guided LCM optimization along a Zigzag camera trajectory.

### 3.3 LAYOUT GENERATION

The first stage of our framework, illustrated in Fig. 2, focuses on generating the layout of the 3D scene through the following three steps:

**LLM-Prompted Layout Initialization.** Given a user description of an indoor scene (e.g., a multi-room house), we leverage Large Language Models (LLMs) (Achiam et al., 2023; Guo et al., 2025) to generate an initial layout that specifies the floorplan of multiple rooms and detailed object configurations, including categories, positions, orientations, and texture descriptions (Fu et al., 2024).

**Iterative Programmatic Verification.** Since layouts generated by this direct prompting approach may exhibit issues such as blank regions or overlapping objects, we introduce an iterative programmatic verification mechanism. Specifically, the 3D layout is converted into executable programs for conflict detection. Identified errors are then fed back into the LLM along with the program for refinement, and this process is repeated until a valid, conflict-free layout is obtained.

**Cluster-Based Orientation Assignment.** To address orientation ambiguities, we introduce a cluster-based strategy in which scene objects are first grouped into functional clusters by LLMs. Within each cluster, object orientations are inferred from inter-object spatial relations (e.g., sofa → TV) rather than relying directly on cardinal directions (e.g., north/south). Once these relative relations are established, they can be straightforwardly converted into cardinal orientations.

### 3.4 SCENE OBJECT GENERATION

The second stage focuses on generating 3D object models from the textual descriptions obtained during layout generation. We adopt 3D Gaussians (Kerbl et al., 2023) as the differentiable representation of objects, owing to their efficiency, explicitness, and compositional flexibility. To optimize these representations, we employ Latent Consistency Models (LCMs) (Luo et al., 2023) together with our proposed Consistency Trajectory Sampling loss ($L_{CTS}$ in Eq. (4)), which enforces both self-consistency and cross-consistency along the sampling trajectory, thereby enabling rapid and faithful text-to-3D synthesis.

Fig. 3 illustrates the optimization of 3D Gaussian object models under $L_{CTS}$. At each optimization step, a rendered view of the 3D Gaussian is fed into a pre-trained diffusion model $\epsilon_\theta$ to generate two adjacent samples along the ODE trajectory. Consistency is then enforced between these samples, effectively distilling deterministic priors into the 3D representation. The overall optimization objective is given by

$$L_{total} = L_{CTS} + L_{scale} + L_{layout} + L_{normal}, \qquad (5)$$

where $L_{scale}$, $L_{layout}$, and $L_{normal}$ serve as auxiliary losses that regularize object scale, spatial placement, and geometry, respectively. Detailed formulations and the complete generation algorithm are provided in the Appendix.

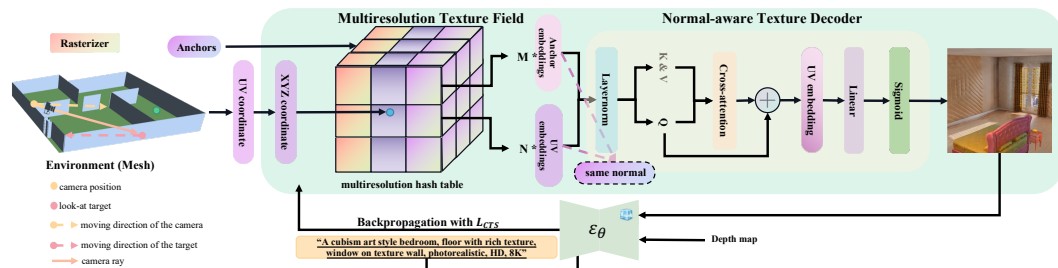

**Figure 4: An illustration of Environment Optimization.** The environment is textured and optimized using a Normal-Aware Texture Field Module under CTS-guided LCM optimization along a Zigzag camera trajectory.

We observe that once the 3D Gaussian representation acquires clear semantic signals aligned with text prompts, injecting noise encourages variations that refine fine-grained alignment. However, since most textual descriptions lack high-frequency details, the objective should shift from strict alignment to detail enhancement. To this end, we adopt a two-stage optimization scheme: in the first stage, noise is added to the rendered image and both terms of $L_{CTS}$ are applied; in the second stage, $x_s$ is set as the noise-free rendered image and only term1 of $L_{CTS}$ is used, allowing the model to focus on detail synthesis while improving training efficiency. The effectiveness of this scheme is supported by our experimental results.

## 3.5 ENVIRONMENT OPTIMIZATION

After generating 3D scene object models, we place them into the empty environment planned from the 3D layout produced in the first stage and proceed to colorize the environment using LCM with our proposed CTS loss. The process consists of three main steps, as illustrated in Fig. 4:

**Camera Pose Generation with Zigzag Trajectory.** For complex multi-room scenes with varying scales, a robust camera trajectory is essential to cover all areas. Existing approaches either rely on pre-defined trajectories (Yang et al., 2024a), which require manual adjustment for different room sizes, or use spherical trajectories (Chen et al., 2024a; Li et al., 2024a), which fail to accommodate all rooms. To address this, we propose a Zigzag camera trajectory that can be automatically generated and adapts to arbitrary room sizes, with the camera and target closely following walls. Specifically, the camera's xy coordinates move opposite to the target's motion, while its height is inversely proportional to the target's height. This design provides three advantages: (1) prevents diffusion models from failing to recognize scene elements at short distances; (2) maintains favorable angles between camera rays and surface normals; (3) maximizes object coverage, reducing texture monotony in rendered views.

**Image Rendering with Normal-Aware Texture Field Module.** Given the sampled camera poses, scene images are obtained for optimization by blending the rendered environment with scene object RGB-D images according to their depth values, treating pixels with smaller depth as foreground. Scene objects are rendered from the optimized 3D Gaussians and composited with the environment to enforce style consistency. The environment is represented as a mesh, augmented with a Normal-Aware Texture Field Module that models learnable textures while mitigating patch artifacts and multi-view inconsistencies, following the design of SceneTex (Chen et al., 2024a).

As in (Chen et al., 2024a), the Normal-Aware Texture Field Module comprises a multi-resolution latent texture field and a texture decoder that maps embeddings to RGB values. Since the environment primarily consists of large planes such as ceilings, floors, and walls, consistent normals generally correspond to consistent texture styles. To leverage this, normals are incorporated into the decoder. For each rasterized UV coordinate, a UV mask isolates coordinates sharing the same normal to form reference embeddings. A multi-head cross-attention module then integrates its rendered UV embedding (Query) with these reference embeddings (Key and Value), followed by an MLP that projects the enhanced embedding to RGB values.

**Environment Optimization with LCM.** Using the scene images rendered along the Zigzag camera trajectory, we employ an LCM as a critic to optimize the texture field, following the SceneTex strategy (Chen et al., 2024a). Optimization is performed with a pre-trained, frozen, depth-conditioned

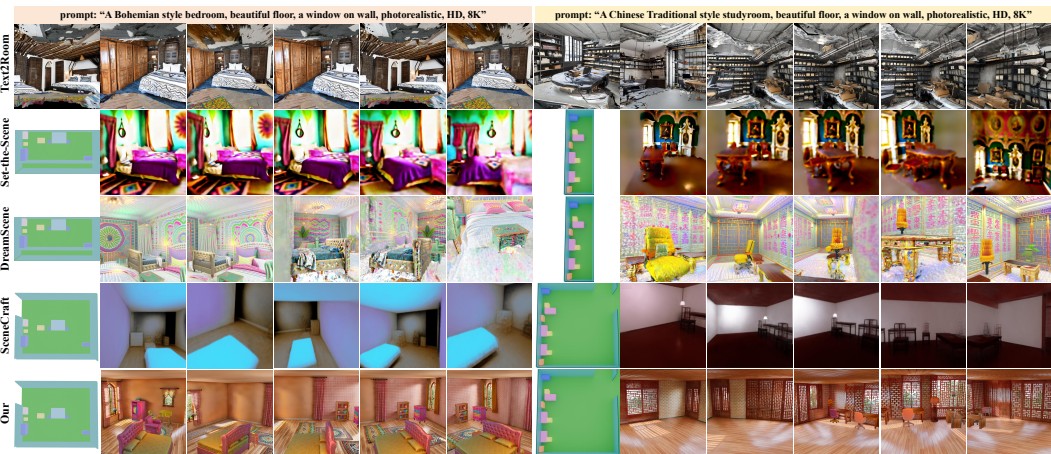

**Figure 5:** Qualitative comparisons of different methods on single-room generation.

diffusion model $\epsilon_\theta$ under the proposed CTS objective $L_{CTS}$ until convergence. In practice, similar to scene object generation, we adopt a two-stage optimization scheme when applying $L_{CTS}$.

## 4 EXPERIMENTS

For the implementation of SceneLCM, we employ GPT-4 (Achiam et al., 2023) for 3D layout generation, Point-E (Nichol et al., 2022) to initialize scene object representations, and the Latent Consistency Model (Luo et al., 2023) for optimizing both scene objects and the environment. All experiments across different models are conducted on a single A800 GPU to ensure fair comparison.

### 4.1 QUALITATIVE RESULTS

**Single-Room Results.** For single-room generation, we compare **SceneLCM** with several open-source methods, including Text2Room (Höllein et al., 2023), Set-the-Scene (Cohen-Bar et al., 2023), SceneCraft (Yang et al., 2024a), and DreamScene (Li et al., 2024a), with qualitative results shown in Fig. 5. Text2Room, which relies on text-conditioned inpainting without explicit 3D layouts, produces scenes that are difficult to control or interact with. Set-the-Scene, based on NeRF composition, cannot handle objects at varying scales, leading to blurred results. SceneCraft finetunes a 2D diffusion model conditioned on layout images with an SDS loss, but struggles with layouts containing many closely placed objects. DreamScene employs Formation Pattern Sampling and a progressive camera strategy to integrate objects and environments, yet the absence of geometric constraints often causes multi-view inconsistencies (e.g., the fused floor and bed in Fig. 5) and artifacts from uncontrolled Gaussian proliferation. In contrast, our model generates complex and realistic scenes while ensuring multi-view consistency.

**Multi-Room Results.** **SceneLCM** is capable of generating multi-room scenes in a fully automatic pipeline from user descriptions, whereas most existing methods are limited to single-room generation. The only exception is SceneCraft (Yang et al., 2024a), which relies on a finetuned diffusion model conditioned on layout images. As shown in Fig. 6, we present qualitative comparisons with SceneCraft using the same layout and camera trajectory for fairness. Our results exhibit more realistic appearance and better alignment with the given layouts, while SceneCraft is constrained by the limited generative capacity of the finetuned diffusion model. Additional multi-room results of SceneLCM are provided in the Appendix.

**Scene Editing Results.** The pipeline of **SceneLCM** supports both texture editing and physically plausible editing. As shown in Fig. 7(a), the first row demonstrates texture editing under four different text conditions for environment optimization, while the second row illustrates physically plausible editing, where a chair initially placed in mid-air falls naturally under gravity, simulated using chair Gaussians (converted to mesh) within the 3D environment.

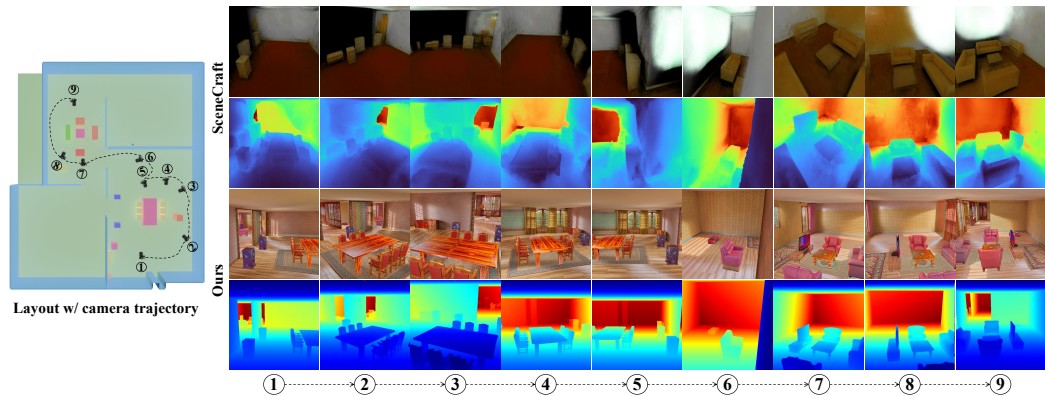

**Figure 6:** Qualitative comparison between SceneCraft (Yang et al., 2024a) and our **SceneLCM** on multi-room generation using the same layout and camera trajectory. Results are shown in order along the trajectory.

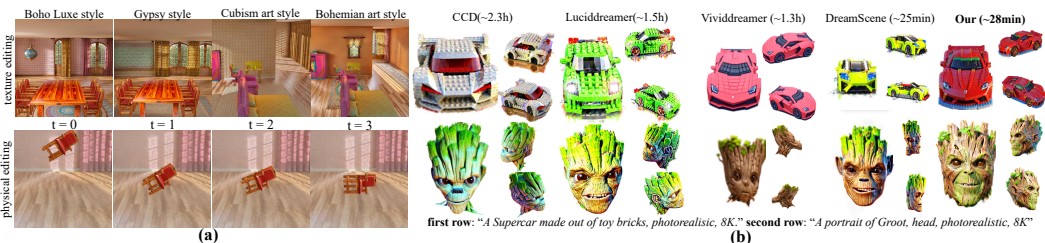

**Figure 7:** (a) **Scene Editing Results.** SceneLCM supports both texture editing and physically plausible editing. (b) **Object Generation Results.** Compared to other methods, SceneLCM produces more realistic 3D objects while achieving faster generation.

**Object Generation Results.** We further compare the object generation performance of **SceneLCM** against existing methods, including CCD (Li et al., 2024b), Luciddreamer (Liang et al., 2024), Vividreamer (Chen et al., 2024b), and DreamScene (Li et al., 2024a). As shown in Fig. 7(b), our method produces 3D objects with more realistic appearances that closely follow text prompts, while achieving faster generation, owing to the LCM under the proposed CTS loss.

## 4.2 QUANTITATIVE RESULTS

In Table 1, we report quantitative results for single-room generation, as most existing methods cannot handle multi-room settings. We randomly choose 8 prompts and 8 rooms for evaluation. We evaluate performance using several metrics: CLIP Score (Hessel et al., 2021) measures the similarity between rendered images and text prompts with CLIP (ViT-B); VQA Score (Lin et al., 2024) evaluates alignment between images and prompts using a VQA model; Inception Score (Yang et al., 2024a) assesses both quality and diversity of generated images. For temporal consistency, given two adjacent RGB-D frames, we warp one frame to another using depth maps and optical flow estimates (Ilg et al., 2017), then compare the errors between them. Additionally, a user study with 20 participants rates the visual quality of 15-second videos (three scenes per method) on a 10-point scale. As shown in the table, the results align with the qualitative analysis in Sec. 4.1, confirming that SceneLCM produces more realistic, consistent, and prompt-aligned scenes. In the table, we also compare per-scene environment optimization time, excluding layout and object generation due to large methodological variance.

### ABLATION STUDY

**Effects of Consistency Trajectory Sampling.** The core of SceneLCM is the Latent Consistency Model (LCM) guided by the proposed Consistency Trajectory Sampling (CTS) loss, where both generation quality and convergence speed depend strongly on the sampling strategy. To ensure

**Table 1:** Quantitative results of different methods on single-room generation. For per-scene running time, we report only the environment optimization, as layout and object generation vary significantly across methods.

| Method | CLIP Score ↑ | Consistency ↓ | VAQ Score ↑ | Inception Score ↑ | User Study ↑ | Time (h) ↓ |
|---|---|---|---|---|---|---|
| Set-the-Scene (Cohen-Bar et al., 2023) | 23.33 | 1.69 | 0.44 | 3.51 | 3.2 | 1.6 |
| DreamScene (Li et al., 2024a) | 24.91 | 1.33 | 0.70 | 4.46 | 7.0 | **1.2** |
| SceneCraft (Yang et al., 2024a) | 21.17 | 1.57 | 0.63 | 3.55 | 4.4 | 4.5 |
| SceneLCM (Ours) | **25.46** | **1.14** | **0.81** | **4.89** | **8.4** | 1.2 |

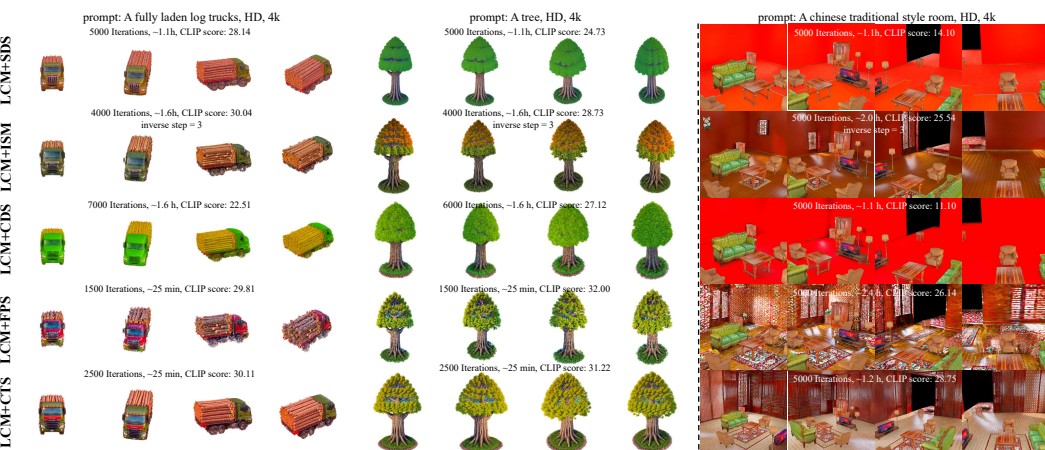

**Figure 8:** Qualitative comparisons of different sampling methods for scene object generation (left) and environment optimization (right). Convergence iterations, running time, and CLIP scores are also reported.

fair comparison, we evaluate different sampling methods combined with CTS on the same LCMs for scene object generation and environment optimization, with qualitative results shown in Fig. 8. Score Distillation Sampling (SDS) (Poole et al., 2022) is the first method to guide optimization by distilling gradients from a pre-trained diffusion model to align the generated output with a target condition, but it suffers from over-smoothing. Interval Score Matching (ISM), proposed in Lucid-Dreamer (Liang et al., 2024), improves SDS by leveraging deterministic diffusion trajectories and interval-based score matching. Consistency Distillation Sampling (CDS), introduced by Consistent3D (Wu et al., 2024), provides more stable and consistent guidance. However, all three methods exhibit slow convergence. DreamScene (Li et al., 2024a) accelerates ISM using Formation Pattern Sampling (FPS), a multi-step 3D object sampling strategy, though the independent noise in each step introduces additional artifacts. In contrast, our Consistency Trajectory Sampling (CTS) ensures self-consistency along the LCM trajectory, achieving both high-quality generation and fast convergence, as confirmed by the qualitative and quantitative results in Fig. 8.

*For more ablation studies upon the pipeline of SceneLCM, please kindly refer to the Appendix!*

## 5 CONCLUSION

We introduced SceneLCM, an automatic indoor scene generation pipeline that integrates LLM-driven layout design with LCM-based optimization through the Consistency Trajectory Sampling (CTS) loss. By combining layout generation, scene object generation, and environment optimization, SceneLCM enables scalable multi-room synthesis, controllable texture editing, and physically coherent modifications. Experiments verify superior quality, efficiency, and editing flexibility over existing approaches, with ablations confirming CTS as a key factor in rapid, high-fidelity generation.

**Limitations.** Although SceneLCM demonstrates strong performance, certain limitations remain. Diffusion model assumptions may cause physically implausible lighting, such as multiple light sources from all windows; inverse rendering could help correct such lighting inconsistencies by explicitly modeling the light transport and environmental context. Additionally, the layout generation process overlooks door placement, occasionally producing obstructive furniture arrangements, which future work will address through door-aware layout planning.

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

Junzhe Zhu, Peiye Zhuang, and Sanmi Koyejo. Hifa: High-fidelity text-to-3d generation with advanced diffusion guidance. In The Twelfth International Conference on Learning Representations.

CONTENTS

## A   THE USE OF LARGE LANGUAGE MODELS (LLMs)

Large language models (LLMs) were used solely as a general-purpose writing aid, such as improving grammar, clarity, and style. All research ideas, experimental designs, implementations, analyses, and scientific claims presented in this paper originate entirely from the authors' discussions. The LLM did not contribute to the conception of the research, the development of methods, or the interpretation of results. The authors take full responsibility for the content of this paper.

Separately, within our proposed approach, LLMs are employed as part of the scene generation pipeline to interpret user descriptions and generate initial layouts. This methodological usage is distinct from the writing assistance described above.

## B   MORE IMPLEMENT DETAILS ABOUT SCENE OBJECT GENERATION

### B.1   ALGORITHM

The process of scene object generation is outlined in Algorithm 1 for clarity and ease of understanding.

### B.2   ADDITIONAL LOSS TERMS

For scene object generation, we incorporate $L_{\text{scale}}$, $L_{\text{layout}}$, and $L_{\text{normal}}$ in addition to the proposed CTS loss for optimization, formulated as follows:

$$
\begin{aligned}
L_{\text{scale}} =& \frac{1}{N} \sum_{i=1}^{N} \max(s_i), \\
L_{\text{normal}} =& \frac{1}{M} \sum_{i=1}^{M} w_i \big(1 - n_i^\top \tilde{n}_i\big), \\
L_{\text{layout}} =& d^x(G_x, x_i, h_i) + d^y(G_y, y_i, w_i) + d^z(G_z, z_i, l_i), \\
d^x(G_x, x_i, h_i) =& \big\| \min(G_x) - \big(x_i - \tfrac{h_i}{2}\big) \big\|_2^2 + \big\| \max(G_x) - \big(x_i + \tfrac{h_i}{2}\big) \big\|_2^2, \\
w_i =& \alpha_i \prod_{j=1}^{i} \big(1 - \alpha_j\big).
\end{aligned}
\tag{6}
$$

Here, $\alpha_i$ denotes the pixel translucency determined by the opacity of the $i$-th Gaussian kernel and the pixel's position. $s_i \in \mathbb{R}^3$ is the scale of the $i$-th Gaussian, $N$ is the number of Gaussians, and $M$ is the number of pixels. Different from Gala3D (Zhou et al., 2024), which enforces objects to strictly lie within bounding boxes, our $L_{\text{layout}}$ encourages tight alignment with box boundaries, thereby enabling direct physics simulation using the bounding boxes.

## C   MORE ABLATION STUDIES

### C.1   LAYOUT GENERATION

**Effects of Iterative Programmatic Verification.**   We compare SceneLCM with other LLM-prompted layout generation methods to highlight the effectiveness of our Iterative Programmatic Verification, which is absent in prior work. As shown in Fig. 9, AnyHome (Fu et al., 2024) often produces unreasonable layouts due to missing furniture, while Holodeck (Yang et al., 2024b) fails to create inter-room passageways and tends to align objects rigidly along walls. InstructScene (Lin & Mu, 2024) and Architect (Wang et al., 2024b) are limited to single-room settings with overly simplistic layouts. In contrast, our approach produces conflict-free and realistic layouts through Iterative Programmatic Verification.

**Effects of Cluster-based Orientation Assignment.**   As shown in Figure 10, our orientation assignment strategy can effectively correct some inaccurate orientations via prompting LLMs for spatial reasoning.

---

**Algorithm 1** Scene Object Generation of SceneLCM

---

**Require:** 3D model parameters $\theta$; training iterations $N$; latent consistency model $\phi$; denoising timesteps from $N_{\min}$ to $N_{\max}$; text prompt $y$; fixed noise $\epsilon$; warm-up timesteps $T_{\text{warm}}$; warm-up steps $N_{\text{warm}}$; cutoff timestep $t_{\text{cut}}$; DPM-Solver $G(\cdot; \cdot, \cdot)$

1: **for** $i = 1, 2, \ldots, N$ **do**
2:     $r \leftarrow 1 - \min(i/N_{\text{warm}}, 1)$
3:     $T_{\min} \leftarrow \text{int}(N_{\min} + r \cdot N_{\text{warm}}), \quad T_{\max} \leftarrow \text{int}(N_{\max} + r \cdot N_{\text{warm}})$
4:     Sample camera pose $c$, compute $x_\pi = g(\theta, c)$
5:     Sample $t \sim [T_{\min}, T_{\max}]$
6:     Sample $s \sim [t - 2t_{\text{cut}}, t - t_{\text{cut}}]$                  ▷ Obtain timesteps $s$ and $t$
7:     **if** $i \leq T_{\text{warm}}$ **then**
8:         $x_s \leftarrow \alpha_s x_\pi + \sigma_s \epsilon$                        ▷ Add noise via DDPM
9:     **else**
10:        $x_s \leftarrow x_\pi$
11:     **end if**
12:    $x_{s \to t} \leftarrow G(x_s; s, t)$                           ▷ Forward Euler solver
13:    **if** $i \leq T_{\text{warm}}$ **then**
14:        $\nabla_\theta L_{\text{CTS}} \leftarrow w_2(s, t)\big(\epsilon_\phi(x_s, s, y) - \epsilon\big)\dfrac{\partial g(\theta, c)}{\partial \theta}$
15:    **else**
16:        $\nabla_\theta L_{\text{CTS}} \leftarrow 0$
17:    **end if**
18:    $\nabla_\theta L_{\text{CTS}} \leftarrow \nabla_\theta L_{\text{CTS}} + w_1(t)\big(\epsilon_\phi(x_t, t, y) - \epsilon_\phi(x_s, s, \emptyset)\big)\dfrac{\partial g(\theta, c)}{\partial \theta}$
19:    $\theta \leftarrow \theta + \eta \nabla_\theta L_{\text{CTS}}$     ▷ Other losses $L_{\text{scale}}$, $L_{\text{layout}}$, and $L_{\text{normal}}$ are omitted here for clarity
20: **end for**

---

## C.2   Scene Object Generation

**Effects of Two-Stage Optimization Scheme.** Once the 3D Gaussian representation captures clear semantic signals aligned with the text prompts, injecting noise can introduce variations that help refine fine-grained alignment. However, as most textual descriptions lack high-frequency details, the optimization objective should shift from strict alignment to enhancing visual detail. To this end, we propose a two-stage optimization scheme; in the second stage, we optimize on noise-free rendered images and remove the second term of our CTS loss. An ablation study on noise removal in the second stage is presented in Fig. 11. Noise removal yields textures with richer details and accelerates generation. Specifically, achieving comparable quality requires 4,000 iterations ( 36 minutes) with added noise, whereas denoising attains similar results within 2,500–3,000 iterations ( 25 minutes).

## C.3   Environment Optimization

**Effects of Zigzag Camera Trajectory** We compare the sphere camera trajectory (Chen et al., 2024a; Li et al., 2024a) with our proposed Zigzag camera trajectory in Fig. 12. The results verify that the Zigzag trajectory offers three advantages: (1) prevents diffusion models from failing to recognize scene elements at close range; (2) maintains favorable angles between camera rays and surface normals; (3) maximizes object coverage, reducing texture monotony in rendered views.

**Effects of Multi-resolution Texture Field.** We conduct ablation experiments on the multi-resolution texture field. As shown in Fig. 13, directly optimizing an RGB texture without the proposed texture field produces highly noisy and unrealistic textures. Moreover, the rapid optimization of the texture map causes multi-view inconsistencies.

**Effects of Normal-aware Texture Decoder** Since the environment mainly consists of large planes such as ceilings, floors, and walls, consistent normals typically correspond to consistent texture styles. To exploit this, normals are incorporated into the texture decoder during Environment Optimization. Qualitative comparisons in Fig. 14 validate the effectiveness of this design.

**Effects of Two-Stage Optimization Scheme.** Similar to Scene Object Generation, we adopt a two-stage optimization scheme in Environment Optimization. Qualitative comparisons in Fig. 15 show that applying the second stage with noise removal produces environments with richer and more detailed textures.

# D    MORE EXPERIMENTAL RESULTS

## D.1    MORE QUANTITATIVE RESULTS

**Layout Functionality.**    For evaluating layout functionality, we adopt the metrics used in Chat2Layout Wang et al. (2024a):

- **OOB**: percentage of layout where objects extend beyond room boundaries or intersect with other objects.
- **ORI**: percentage of correctly oriented objects.
- **FFR**: furniture footprint ratio, i.e., the floor area occupied by furniture divided by the area of the room.

We evaluate these metrics on 10 rooms, following Chat2Layout. Each layout is manually checked, and the values are reported. Notably, OOB is 0 for AnyHome (Fu et al., 2024), Holodeck (Yang et al., 2024b), Architect (Wang et al., 2024b), and our method. Specifically, AnyHome discards overlapping or out-of-bound furniture, Holodeck and Architect rerun the solver, and our method employs Iterative Programmatic Verification until error-free. These mechanisms guarantee OOB = 0 at the code level, which our experiments confirm.

As shown in Tab. 2, InstructScene achieves the highest ORI in single-room settings, while our method achieves the highest ORI in multi-room settings. For FFR, InstructScene first generates furniture and then constructs a minimal floorplan enclosing it; hence, its FFR is not reported. Our method outperforms all baselines in terms of FFR.

## D.2    MORE QUALITATIVE RESULTS

**More Physical Editing Results.** See Fig. 16.

**More Object Generation Results.** See furniture results in Fig. 17 and other object results in Fig. 18.

**More Multi-Room Results.**  See Fig. 19–22.

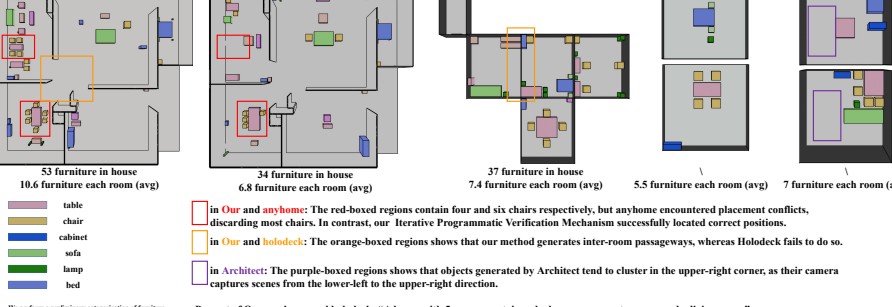

**Figure 9:** Quantitative comparisons of different methods for layout generation.

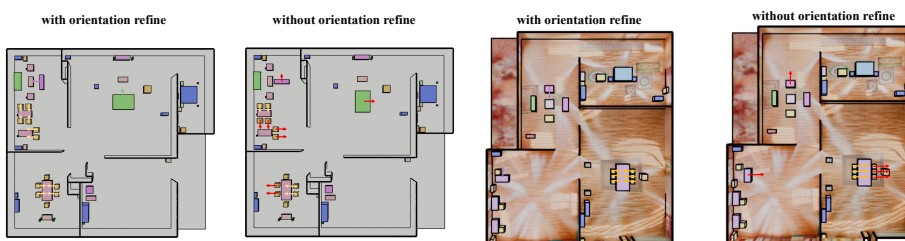

**Figure 10:** Qualitative comparison of SceneLCM with and without Cluster-based Orientation Assignment. Red arrows highlight incorrect orientations, as LLMs often produce uniform orientations for identical objects. Our method effectively rectifies these errors.

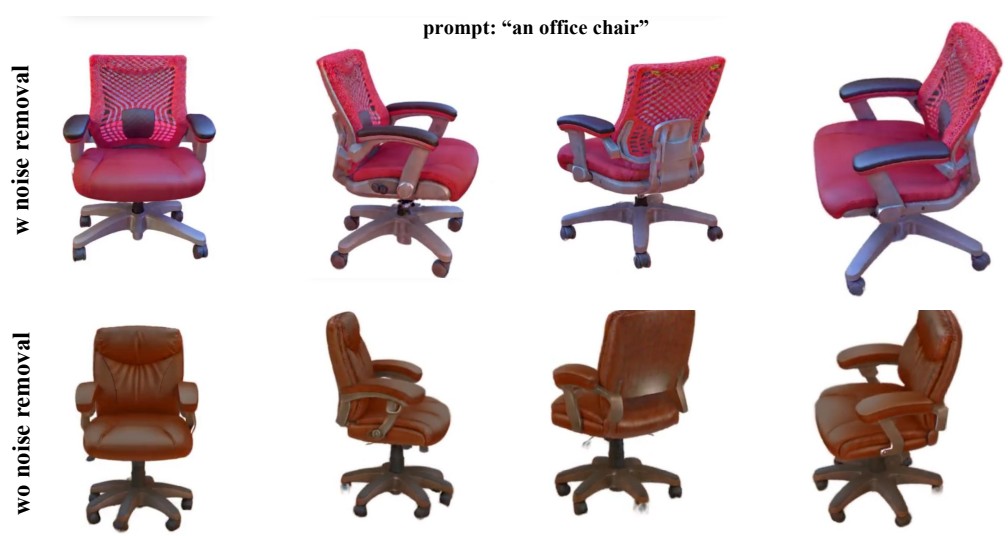

**Figure 11:** Qualitative results of noise removal in the second stage for Scene Object Generation.

## E  THEORETICAL PROOF

We provide two theoretical justifications: (i) the proposed CTS loss is mathematically equivalent to the Consistency Loss (Song et al., 2023), and (ii) upon convergence, it ensures the generation of high-fidelity 3D models. These results correspond to Theorem 1 and Theorem 2.

**Theorem 1** *Let $f_\theta(\cdot)(x, t)$ denote the pre-trained consistency function. We assume $f_\theta(\cdot)$ satisfies the formulation defined in Latent Consistency ModelLuo et al. (2023), and $t \geq 30$. Assume further that for all $t \geq 30$, the ODE solver $G$ called at $t_{n+1}$ has local error uniformly bounded by $O((t_{n+1} - t_n)^{p+1})$ with $p \geq 1$, The Consistency LossLuo et al. (2023) can be mathematically expressed as the sum of the Consistency Trajectory Sampling Loss and two infinitesimal components, along with a term whose magnitude is bounded by $10^{-7}$:*

$$L_{CD} = L_{CTS} + \underbrace{(-\frac{O(h^2)}{\alpha_{t_{n+1}}}) + c_{skip}(t_{n+1})O((\triangle t)^{p+1})}_{\textit{infinitesimal components}} + \underbrace{m}_{|m| \leq 10^{-7}}$$

$$= o((\triangle t)^2) + m$$

$$h = log(\frac{\alpha_{t_n}}{\sigma_{t_n}}) - log(\frac{\alpha_{t_{n+1}}}{\sigma_{t_{n+1}}})$$

(7)

**Proof 1** *The proof is based on the formulation defined in Latent Consistency Model (Luo et al., 2023). We have $f_\theta(x, t) = c_{skip}(t)x + c_{out}(t)F_\theta(x_t, t)$, where $F_\theta(x_t, t) = \frac{x_t - \sigma_t \epsilon_\theta(x_t, t)}{\alpha_t}$,*

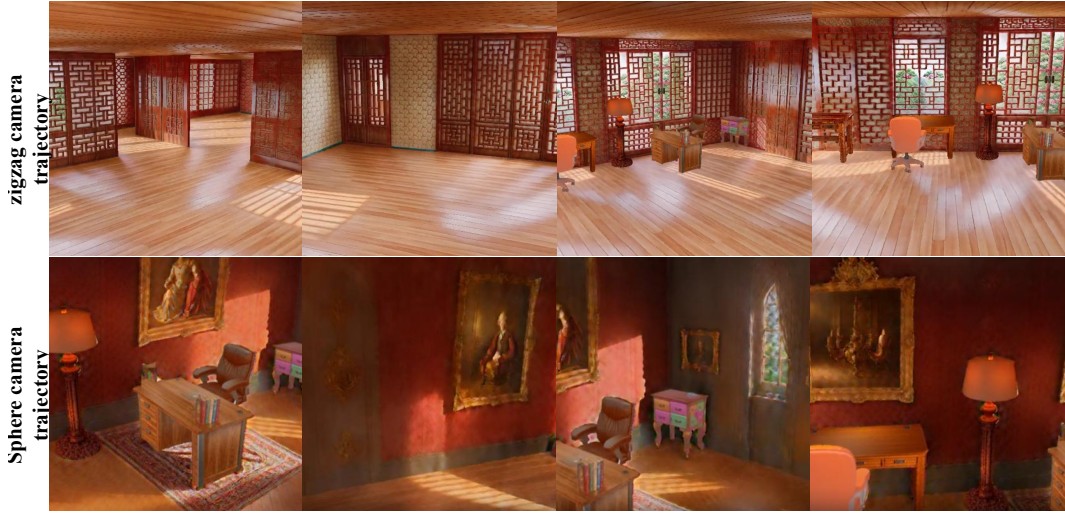

**Figure 12:** Qualitative comparisons between our Zigzag camera trajectory and the Sphere camera trajectory. Using the Sphere trajectory, placing the camera at the center prevents capturing combined entities (wall + floor + ceiling), limiting the model's ability to recognize scene semantics and resulting in blurred textures.

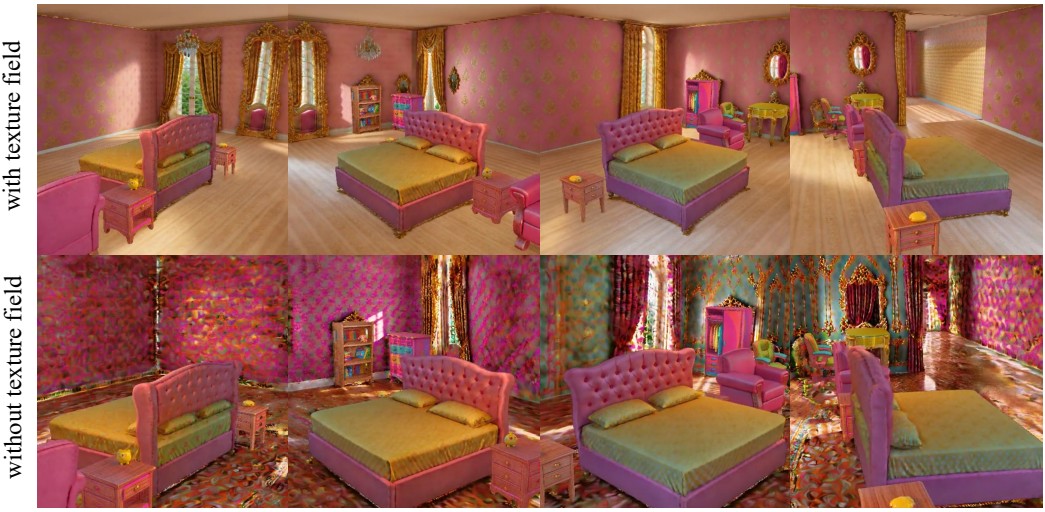

**Figure 13:** Qualitative comparison of multi-resolution texture field and UV texture image. Directly optimizing an RGB texture without the multi-resolution texture field results in severe noise and unrealistic appearances. Moreover, the fast optimization process introduces multi-view inconsistencies.

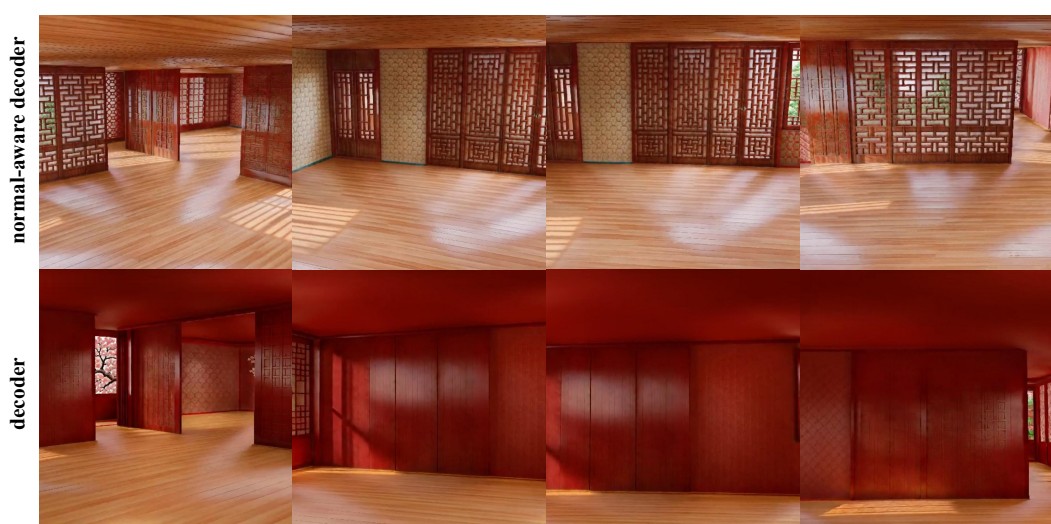

**Figure 14:** Qualitative comparison of texture decoder with and without normal awareness.

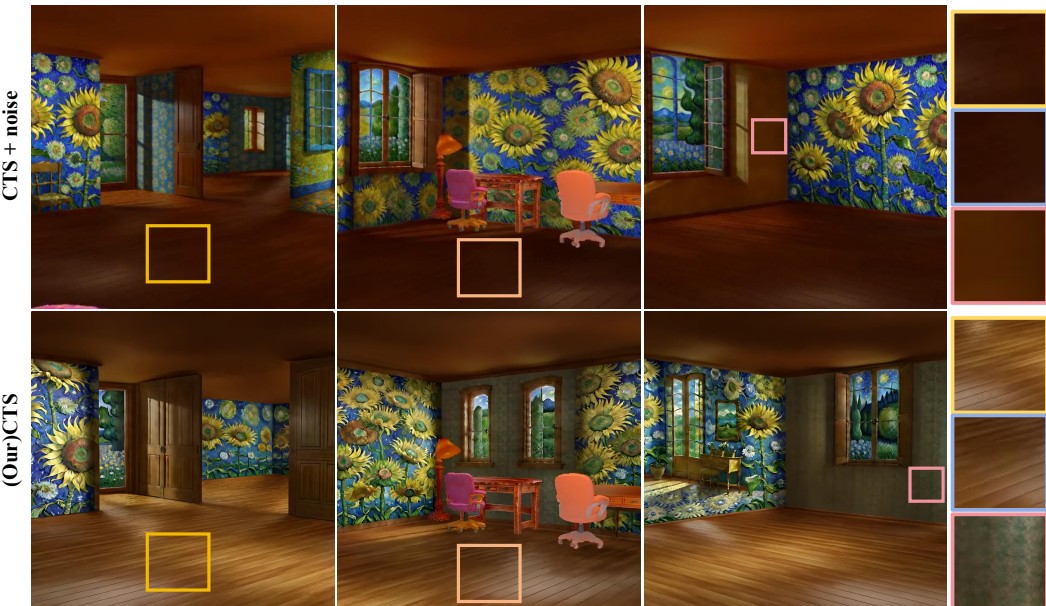

**Figure 15:** Qualitative results of noise removal in the second stage for Environment Optimization.

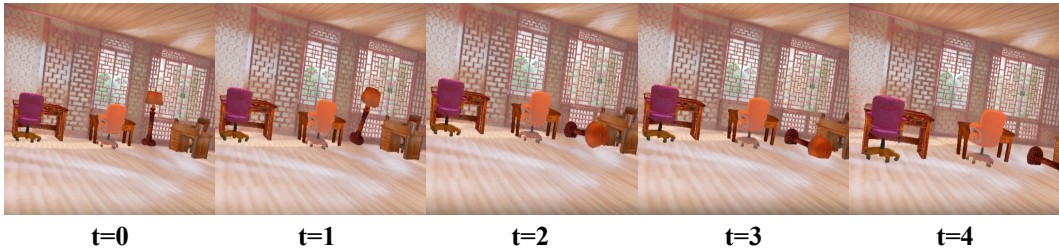

**Figure 16:** Additional physical editing results. Tilting the room by 30° causes objects to slide downward under gravity.

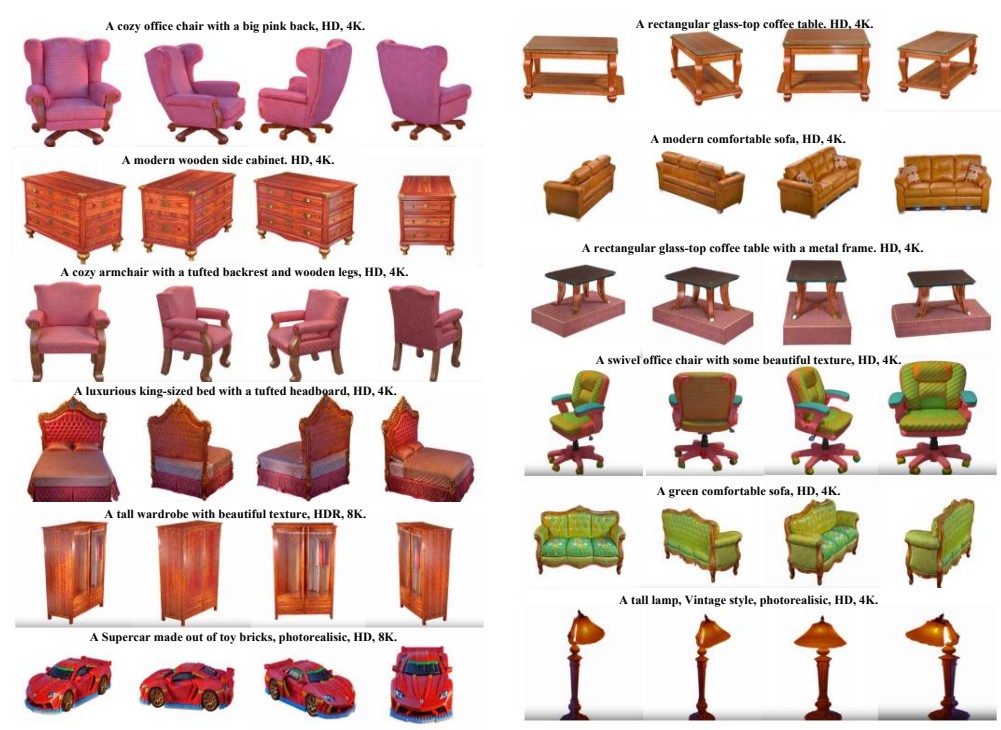

**Figure 17:** More furniture results of scene object generation.

**Table 2:** Quantitative results of layout functionality.

| score | AnyhomeFu et al. (2024) | holodeckYang et al. (2024b) | InstructSceneLin & Mu (2024) | ArchitectWang et al. (2024b) | Our |
|---|---|---|---|---|---|
| OOB ↓ | 0 | 0 | 28.57% | 0 | 0 |
| ORI ↑ | 76.66% | 74.35% | 89.22% | 73.32% | 83.57% |
| FFR ↑ | 7.14% | 9.01% | - | 7.85% | 9.52% |

$c_{skip}(t) = \frac{\sigma^2}{(\frac{t}{0.1})^2+\sigma^2}$, $c_{out}(t) = \frac{\frac{t}{0.1}}{\sqrt{(\frac{t}{0.1})^2+\sigma^2}}$, *and* $\sigma = \frac{1}{2}$. $x_t$ *is obtained by ODE solver applied to* $x_s$. *To simplicity, we omit the condition* $c$ *of the consistency function and exponential moving average(EMA) of the parameter* $\theta$. *We use the DPM-solver as ODE solver* $G(x_s, s, t)$, $x_{s \to t} = G(x_s, s, t)$, *where* $x_{s \to t}$ *is obtained by ODE solver from s to t.*

*For simplicity, the absolute value notation is omitted in following derivation.*

$$
\begin{aligned}
L_{CD} =& ||f_\theta(x_{t_n}, t_n) - f_\theta(x_{t_n \to t_{n+1}}, t_{n+1})||_2^2 \\
=& c_{skip}(t_n)x_{t_n} + c_{out}(t_n)F_\theta(x_{t_n}, t_n) - c_{skip}(t_{n+1})x_{t_n \to t_{n+1}} - c_{out}(t_{n+1})F_\theta(x_{t_n \to t_{n+1}}, t_{n+1}) \\
=& \underbrace{[c_{skip}(t_n)x_{t_n} - c_{skip}(t_{n+1})x_{t_n \to t_{n+1}}]}_{\textbf{term 1}} + \underbrace{c_{out}(t_n)[F_\theta(x_{t_n}, t_n) - F_\theta(x_{t_n \to t_{n+1}}, t_{n+1})]}_{\textbf{term 2}} \\
& + \underbrace{[c_{out}(t_n) - c_{out}(t_{n+1})]F_\theta(x_{t_n \to t_{n+1}}, t_{n+1})}_{\textbf{term 3}}
\end{aligned}
$$

(8)

*where* $x_{t_n \to t_{n+1}}$ *is calculated given ODE solver G as* $x_{t_n \to t_{n+1}} = G(x_{t_n}; t_n, t_{n+1})$.

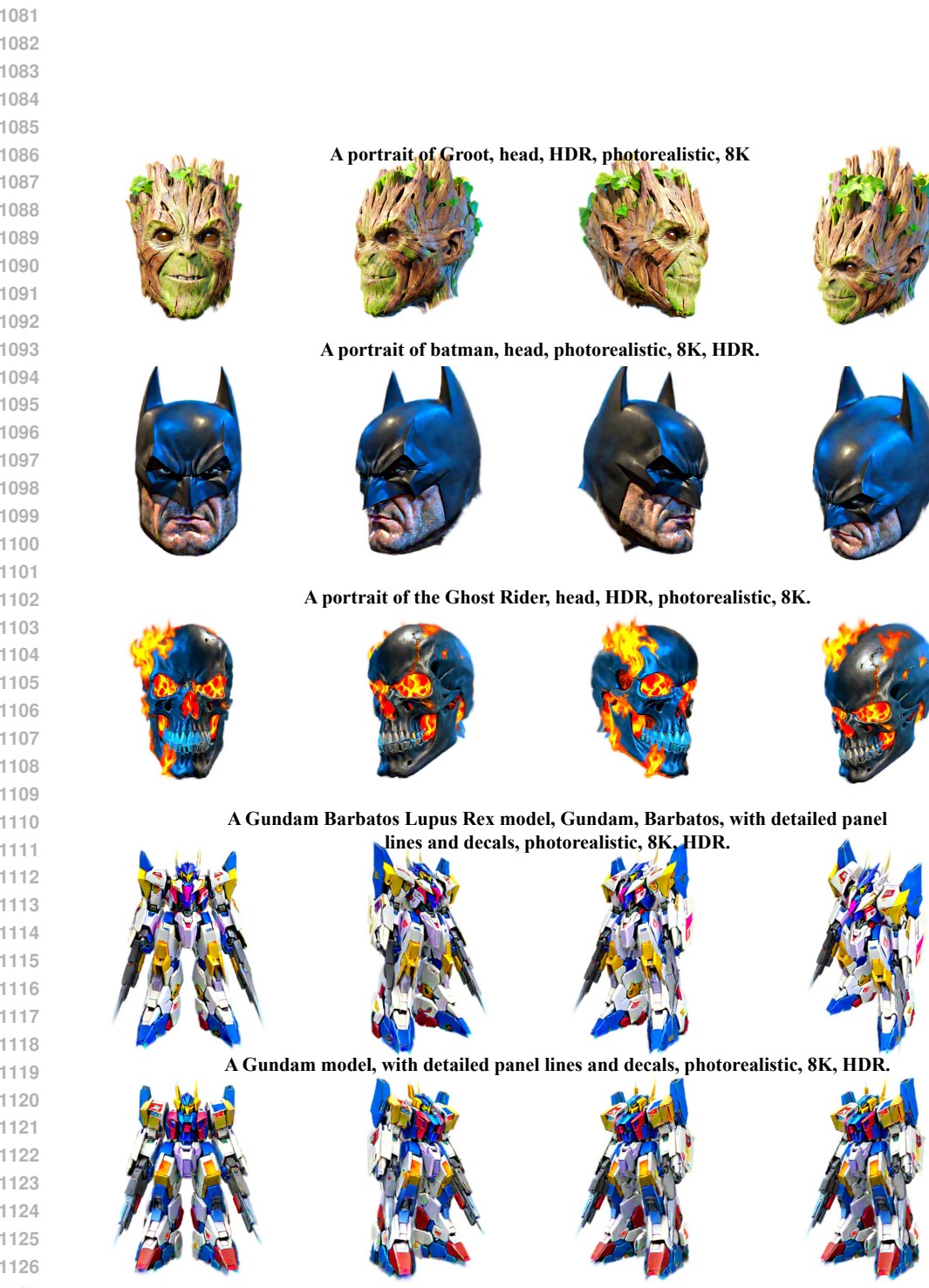

**Figure 18:** More object results of scene object generation.

**Layout 001**

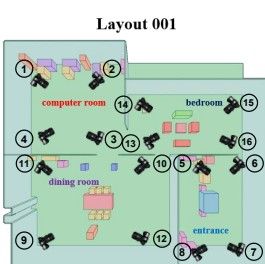

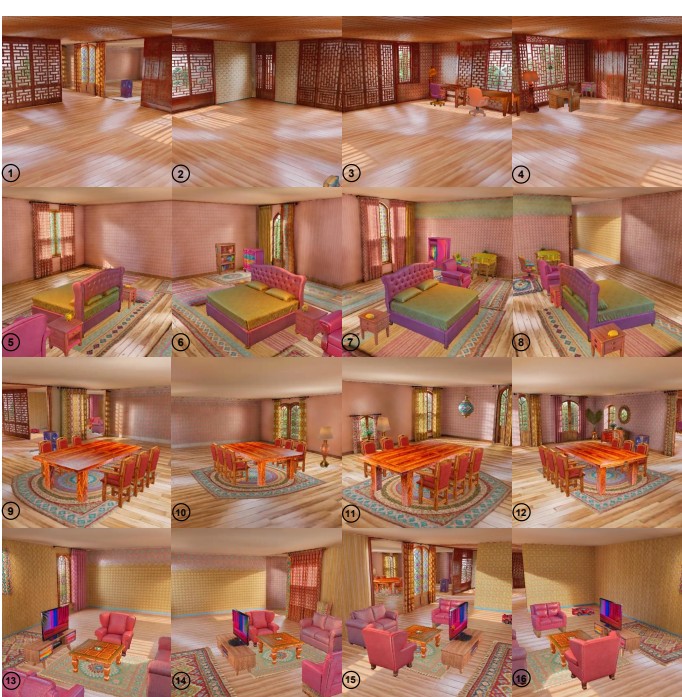

**prompt:** A house contains a computer room, a bedroom and a big dining room. Totally 4 rooms.
**4 room:** computer room, bedroom, dining room, entrance
**37 furniture:** desk, lamp, table, chair, armchair ...
**prompt for each room**(LLM automatically generates prompts for each room and all furniture after parsing the prompt)**:**

- **computer room:** A **traditional chinese** study room, beautiful floor, a window on wall, photorealistic, HD, 8K
- **bedroom:** A **Boho-Hippie** bedroom, beautiful floor, a window on wall, photorealistic, HD, 8K
- **dining room:** A **Gypsy-classic** style dining room, beautiful floor, a window on wall, photorealistic, HD, 8K
- **entrance:** A **Gypsy** style entrance, beautiful floor, a window on wall, photorealistic, HD, 8K

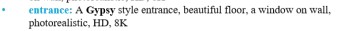

**Figure 19:** More qualitative results on multi-room generation (1).

**Layout 002**

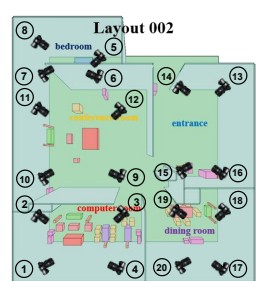

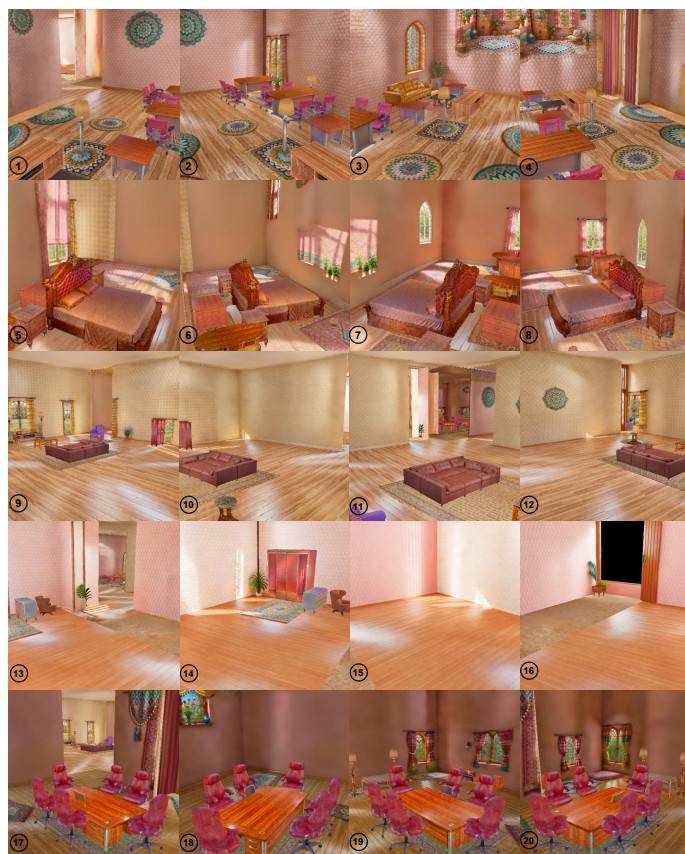

**prompt:** A house with 5 rooms, contains a bedroom, a computer room and a living room. Totally 5 rooms.
**5 room:** computer room, bedroom, living room, entrance, conference room
**59 furniture:** desk, lamp, table, chair, armchair ...
**prompt for each room**(LLM automatically generates prompts for each room and all furniture after parsing the prompt)**:**

- **computer room:** A **Boho Nature** style computer room, beautiful floor, a window on wall, photorealistic, HD, 8K
- **bedroom:** A **Boho-Hippie** bedroom, beautiful floor, a window on wall, photorealistic, HD, 8K
- **living room:** A **Boho Luxe** style dining room, beautiful floor, a window on wall, photorealistic, HD, 8K
- **entrance:** A **modern bright** style entrance, beautiful floor, a window on wall, photorealistic, HD, 8K
- **conference room:** A **Aryan** style conference room, beautiful floor, a window on wall, photorealistic, HD, 8K

**Figure 20:** More qualitative results on multi-room generation (2).

**Layout 003**

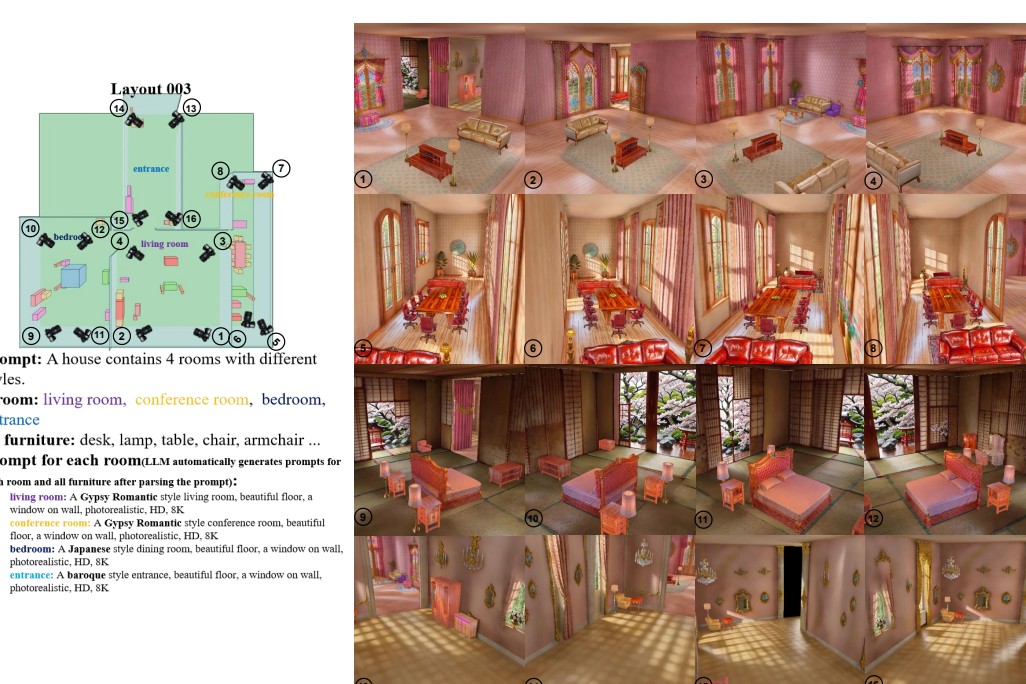

**prompt:** A house contains 4 rooms with different styles.

**4 room:** living room, conference room, bedroom, entrance

**47 furniture:** desk, lamp, table, chair, armchair ...

**prompt for each room**(LLM automatically generates prompts for each room and all furniture after parsing the prompt):

- **living room:** A **Gypsy Romantic** style living room, beautiful floor, a window on wall, photorealistic, HD, 8K
- **conference room:** A **Gypsy Romantic** style conference room, beautiful floor, a window on wall, photorealistic, HD, 8K
- **bedroom:** A **Japanese** style dining room, beautiful floor, a window on wall, photorealistic, HD, 8K
- **entrance:** A **baroque** style entrance, beautiful floor, a window on wall, photorealistic, HD, 8K

**Figure 21:** More qualitative results on multi-room generation (3).

**Layout 004**

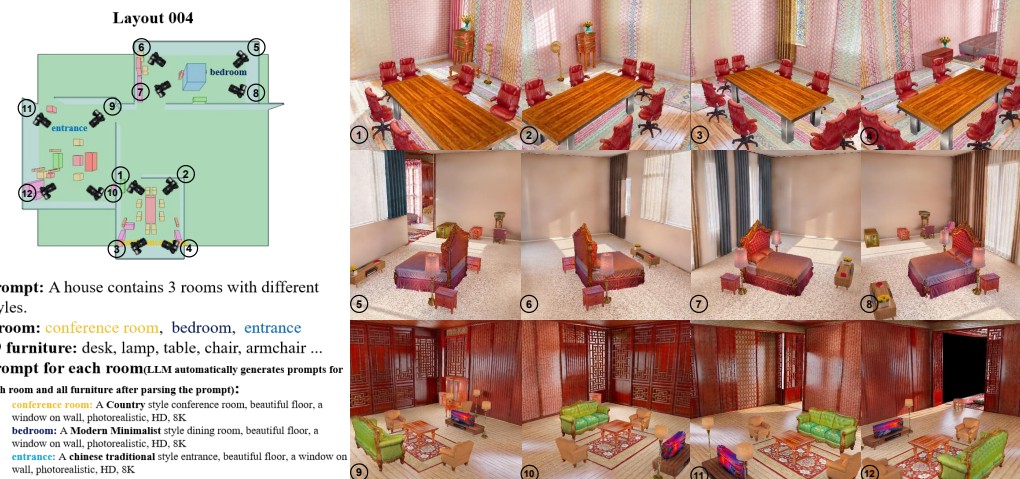

**prompt:** A house contains 3 rooms with different styles.

**3 room:** conference room, bedroom, entrance

**39 furniture:** desk, lamp, table, chair, armchair ...

**prompt for each room**(LLM automatically generates prompts for each room and all furniture after parsing the prompt):

- **conference room:** A **Country** style conference room, beautiful floor, a window on wall, photorealistic, HD, 8K
- **bedroom:** A **Modern Minimalist** style dining room, beautiful floor, a window on wall, photorealistic, HD, 8K
- **entrance:** A **chinese traditional** style entrance, beautiful floor, a window on wall, photorealistic, HD, 8K

**Figure 22:** More qualitative results on multi-room generation (4).

**term 1**:

$$
\begin{aligned}
[c_{skip}(t_n)x_{t_n} - c_{skip}(t_{n+1})x_{t_n \to t_{n+1}}] =& c_{skip}(t_n)x_{t_n} - c_{skip}(t_{n+1})G(x_{t_n}; t_n, t_{n+1}) \\
=& c_{skip}(t_n)x_{t_n} - c_{skip}(t_{n+1})(G(x_{t_n}; t_n, t_{n+1}) - x_{t_{n+1}} + x_{t_{n+1}}) \\
=& [c_{skip}(t_n)x_{t_n} - c_{skip}(t_{n+1})x_{t_{n+1}}] + c_{skip}(t_{n+1})(x_{t_{n+1}} - G(x_{t_n}; t_n, t_{n+1})) \\
\leq& c_{skip}(t_n)(\alpha_{t_n}x_\pi + \sigma_{t_n}\epsilon]) - c_{skip}(t_{n+1})(\alpha_{t_{n+1}}x_\pi + \sigma_{t_{n+1}}\epsilon]) \\
& + c_{skip}(t_{n+1}) \underbrace{O((t_n - t_{n+1})^{p+1})}_{\text{local error of Euler soler}} \\
=& (c_{skip}(t_n)\alpha_{t_n} - c_{skip}(t_{n+1})\alpha_{t_{n+1}})x_\pi \\
& + [c_{skip}(t_n)\sigma_{t_n} - c_{skip}(t_{n+1})\sigma_{t_{n+1}}]\epsilon \\
& + c_{skip}(t_{n+1})O((\triangle t)^{p+1})
\end{aligned}
\tag{9}
$$

where $0 < \alpha_{t_n}, \sigma_{t_n} < 1$. When $t \geq 30$, $c_{skip}(t)$ is monotonically decreasing, and $0 < c_{skip}(t)\alpha_t \leq 10^{-7}$. Because all experiments were conducted under FP16 configuration, $c_{skip}(t)\alpha_t, c_{skip}(t)\sigma_t \approx 0$. Beside, $t$ exceeds 100 in the vast majority of cases. term 1 can be simplified as:

$$
[c_{skip}(t_n)x_{t_n} - c_{skip}(t_{n+1})x_{t_n \to t_{n+1}}] = m + c_{skip}(t_{n+1})O((\triangle t)^{p+1})
\tag{10}
$$

where $m$ is a value so small that it is effectively negligible in devices.

**term 2**: We use the DPM-solver as ODE solver $G(x_s, s, t)$ :

$$
\begin{aligned}
[F_\theta(x_{t_n}, t_n) - F_\theta(x_{t_n \to t_{n+1}}, t_{n+1})] =& \frac{x_{t_n} - \sigma_{t_n}\epsilon_\theta(x_{t_n}, t_n)}{\alpha_{t_n}} - \frac{x_{t_n \to t_{n+1}} - \sigma_{t_{n+1}}\epsilon_\theta(x_{t_n \to t_{n+1}}, t_{n+1})}{\alpha_{t_{n+1}}} \\
=& \frac{x_{t_n} - \sigma_{t_n}\epsilon_\theta(x_{t_n}, t_n)}{\alpha_{t_n}} \\
& - \frac{\frac{\alpha_{t_{n+1}}}{\alpha_{t_n}}x_{t_n} - \sigma_{t_{n+1}}(e^{-h} - 1)\epsilon_\theta(x_{t_n}, t_n) + O(h^2) - \sigma_{t_{n+1}}\epsilon_\theta(x_{t_n \to t_{n+1}}, t_{n+1})}{\alpha_{t_{n+1}}} \\
=& \frac{x_{t_n} - \sigma_{t_n}\epsilon_\theta(x_{t_n}, t_n)}{\alpha_{t_n}} \\
& - \frac{\frac{\alpha_{t_{n+1}}}{\alpha_{t_n}}x_{t_n} - \alpha_{t_{n+1}}(\frac{\sigma_{t_n}}{\alpha_{t_n}} - \frac{\sigma_{t_{n+1}}}{\alpha_{t_{n+1}}})\epsilon_\theta(x_{t_n}, t_n) + O(h^2) - \sigma_{t_{n+1}}\epsilon_\theta(x_{t_n \to t_{n+1}}, t_{n+1})}{\alpha_{t_{n+1}}} \\
=& \frac{x_{t_n} - \sigma_{t_n}\epsilon_\theta(x_{t_n}, t_n)}{\alpha_{t_n}} \\
& - (\frac{x_{t_n}}{\alpha_{t_n}} - \frac{\sigma_{t_n}}{\alpha_{t_n}}\epsilon_\theta(x_{t_n}, t_n) + \frac{\sigma_{t_{n+1}}}{\alpha_{t_{n+1}}}(\epsilon_\theta(x_{t_n}, t_n) - \epsilon_\theta(x_{t_n \to t_{n+1}}, t_{n+1}))) - \frac{O(h^2)}{\alpha_{t_{n+1}}} \\
=& \frac{\sigma_{t_{n+1}}}{\alpha_{t_{n+1}}}(\epsilon_\theta(x_{t_n \to t_{n+1}}, t_{n+1}) - \epsilon_\theta(x_{t_n}, t_n)) - \frac{O(h^2)}{\alpha_{t_{n+1}}}
\end{aligned}
\tag{11}
$$

Finally, term 2 can be simplified as followed:

$$
c_{out}(t_n)[F_\theta(x_{t_n}, t_n) - F_\theta(x_{t_n \to t_{n+1}}, t_{n+1})] = c_{out}(t_n)\frac{\sigma_{t_{n+1}}}{\alpha_{t_{n+1}}}(\epsilon_\theta(x_{t_n \to t_{n+1}}, t_{n+1}) - \epsilon_\theta(x_{t_n}, t_n)) - \frac{O(h^2)}{\alpha_{t_{n+1}}}
\tag{12}
$$

**term 3**:

$$[c_{out}(t_n) - c_{out}(t_{n+1})]F_\theta(x_{t_n \to t_{n+1}}, t_{n+1}) = [c_{out}(t_n) - c_{out}(t_{n+1})](\frac{x_{t_n \to t_{n+1}} - \sigma_{t_{n+1}}\epsilon_\theta(x_{t_n \to t_{n+1}}, t_{n+1})}{\alpha_{t_{n+1}}})$$

$$\overset{(i)}{=} [c_{out}(t_n) - c_{out}(t_{n+1})](\frac{x_{t_n} - \sigma_{t_n}\epsilon_\theta(x_{t_n}, t_n)}{\alpha_{t_n}})$$

$$+ [c_{out}(t_n) - c_{out}(t_{n+1})](\frac{\sigma_{t_{n+1}}}{\alpha_{t_{n+1}}}(\epsilon_\theta(x_{t_n}, t_n) - \epsilon_\theta(x_{t_n \to t_{n+1}}, t_{n+1})))$$

$$= [c_{out}(t_n) - c_{out}(t_{n+1})]x_\pi$$

$$+ [c_{out}(t_n) - c_{out}(t_{n+1})](\frac{\sigma_{t_n}}{\alpha_{t_n}})(\epsilon - \epsilon_\theta(x_{t_n}, t_n))$$

$$+ [c_{out}(t_n) - c_{out}(t_{n+1})](\frac{\sigma_{t_{n+1}}}{\alpha_{t_{n+1}}}(\epsilon_\theta(x_{t_n}, t_n) - \epsilon_\theta(x_{t_n \to t_{n+1}}, t_{n+1})))$$

$$(13)$$

where $(i)$ hold according to the DPM-solver.

In our setting, the time interval is 100 and $t \geq 30$. Therefore, $|c_{out}(t_n) - c_{out}(t_{n+1})| \leq 10^{-7}$. However, in early stage, $\frac{\sigma_{t_n}}{\alpha_{t_n}} \geq 1$ and $[c_{out}(t_n) - c_{out}(t_{n+1})](\frac{\sigma_{t_n}}{\alpha_{t_n}})$ cannot be neglected. As number of iterations increases, the time $t$ will gradually decrease, and $\frac{\sigma_{t_n}}{\alpha_{t_n}} < 1$. At this point, $[c_{out}(t_n) - c_{out}(t_{n+1})](\frac{\sigma_{t_n}}{\alpha_{t_n}})$ can be neglected. Then, term 3 can be formula as:

$$[c_{out}(t_n) - c_{out}(t_{n+1})]F_\theta(x_{t_n \to t_{n+1}}, t_{n+1}) = m + [c_{out}(t_n) - c_{out}(t_{n+1})](\frac{\sigma_{t_n}}{\alpha_{t_n}})(\epsilon - \epsilon_\theta(x_{t_n}, t_n))$$

$$+ [c_{out}(t_n) - c_{out}(t_{n+1})](\frac{\sigma_{t_{n+1}}}{\alpha_{t_{n+1}}}(\epsilon_\theta(x_{t_n}, t_n) - \epsilon_\theta(x_{t_n \to t_{n+1}}, t_{n+1})))$$

$$(14)$$

Finally, combining the results from term 1, term 2, and term 3, we obtain our CTS loss:

$$L_{CD} = ||f_\theta(x_{t_n}, t_n) - f_\theta(x_{t_n \to t_{n+1}}, t_{n+1})||_2^2$$

$$= ||m + c_{skip}(t_{n+1})O((\triangle t)^{p+1})$$

$$+ c_{out}(t_n)\frac{\sigma_{t_{n+1}}}{\alpha_{t_{n+1}}}(\epsilon_\theta(x_{t_n \to t_{n+1}}, t_{n+1}) - \epsilon_\theta(x_{t_n}, t_n)) - \frac{O(h^2)}{\alpha_{t_{n+1}}}$$

$$+ m + [c_{out}(t_n) - c_{out}(t_{n+1})](\frac{\sigma_{t_n}}{\alpha_{t_n}})(\epsilon - \epsilon_\theta(x_{t_n}, t_n))$$

$$+ [c_{out}(t_n) - c_{out}(t_{n+1})](\frac{\sigma_{t_{n+1}}}{\alpha_{t_{n+1}}}(\epsilon_\theta(x_{t_n}, t_n) - \epsilon_\theta(x_{t_n \to t_{n+1}}, t_{n+1})))||_2^2$$

$$\overset{(ii)}{=} ||c_{out}(t_{n+1})(\frac{\sigma_{t_{n+1}}}{\alpha_{t_{n+1}}})(\epsilon_\theta(x_{t_n \to t_{n+1}}, t_{n+1}) - \epsilon_\theta(x_{t_n}, t_n))||_2^2 + ||[c_{out}(t_{n+1}) - c_{out}(t_n)](\frac{\sigma_{t_n}}{\alpha_{t_n}})(\epsilon_\theta(x_{t_n}, t_n) - \epsilon)||_2^2$$

$$- \frac{O(h^2)}{\alpha_{t_{n+1}}} + c_{skip}(t_{n+1})O((\triangle t)^{p+1}) + m$$

$$= L_{CTS} - \frac{O(h^2)}{\alpha_{t_{n+1}}} + c_{skip}(t_{n+1})O((\triangle t)^{p+1}) + m$$

$$(15)$$

where $(ii)$ is because we follow DreamFusion (Poole et al., 2022) and omit the U-Net Jacobian term in practice. $\frac{\sigma_{t_n}}{\alpha_{t_n}} > 1$ in early stage, we can not omit the second term.

Therefore, we obtain two conclusions:

- $L_{CTS}$ can be expressed as the sum the $L_{CD}$ and two infinitesimal components, along with a term whose magnitude is bounded by $10^{-7}$.

$$L_{CD} = L_{CTS} + \underbrace{(-\frac{O(h^2)}{\alpha_{t_{n+1}}}) + c_{skip}(t_{n+1})O((\triangle t)^{p+1})}_{\text{infinitesimal components}} + \underbrace{m}_{|m| \leq 10^{-7}} \qquad (16)$$

$$h = log(\frac{\alpha_{t_n}}{\sigma_{t_n}}) - log(\frac{\alpha_{t_{n+1}}}{\sigma_{t_{n+1}}})$$

- $L_{CTS}$ can formula as:

$$L_{CTS} = ||w_1(\epsilon_\theta(x_{t_n \to t_{n+1}}, t_{n+1}) - \epsilon_\theta(x_{t_n}, t_n))||_2^2 + ||w_2(\epsilon_\theta(x_{t_n}, t_n) - \epsilon)||_2^2$$

$$w_1 = c_{out}(t_{n+1})(\frac{\sigma_{t_{n+1}}}{\alpha_{t_{n+1}}}) \qquad (17)$$

$$w_2 = [c_{out}(t_{n+1}) - c_{out}(t_n)](\frac{\sigma_{t_n}}{\alpha_{t_n}})$$

*The proof is completed.*

**Theorem 2** *Assume that the pre-trained noise predictor $\epsilon_\theta(\cdot; \cdot)$ satisfies the Lipschitz condition. Define $\triangle := sup|t_1 - t_2)|$. For any given camera pose $\pi$, if convergence is achieved according to $L_{CTS}$, then there exists a corresponding real image $x_0 \sim p_{data}(x)$ such that:*

$$||x_\pi - x_0||_2 = O(\triangle) \qquad (18)$$

*where $x_\pi = g(\theta, \pi)$ denotes the rendered image for pose $\pi$.*

*We offer two ways of proof. For the **Proof 1**, we directly utilize the CTS loss. For the **Proof 2**, we make use of Theorem 1.*

***Proof 1:***

***Proof 2*** *Given $L_{CTS}(\xi) = 0$, for any $t$, $s$, we have $\epsilon(x_{s \to t}, t) = \epsilon_\theta(x_s, s)$ and $T \geq t_n \geq t_{n-1} \geq 0$. Assume $G$ is DPM solver and follow the first-order definition of DPM-Solver, given $e$, we have:*

$$\begin{aligned}
G(x_{s \to t}, t, e) &= \frac{\alpha_e}{\alpha_t}x_{s \to t} - \sigma_e(e^{-h} - 1)\epsilon_\theta(x_{s \to t}, t) \\
&= \frac{\alpha_e}{\alpha_t}x_{s \to t} - \alpha_e(\frac{\sigma_t}{\alpha_t} - \frac{\sigma_e}{\alpha_e})\epsilon_\theta(x_{s \to t}, t) \\
&= \alpha_e \frac{x_{s \to t} - \sigma_t\epsilon_\theta(x_{s \to t}, t)}{\alpha_t} + \sigma_e\epsilon_\theta(x_{s \to t}, t) \\
&= \alpha_e \frac{\alpha_t(\frac{x_s - \sigma_s\epsilon_\theta(x_s, s)}{\alpha_s}) + \sigma_t\epsilon(x_s, s) - \sigma_t\epsilon_\theta(x_{s \to t}, t) + O(h^2)}{\alpha_t} + \sigma_e\epsilon_\theta(x_{s \to t}, t) \\
&\overset{(iii)}{=} \alpha_e(\frac{x_s - \sigma_s\epsilon_\theta(x_s, s)}{\alpha_s}) + \sigma_e\epsilon_\theta(x_s, s) + L_1 O(h^2) \\
&= G(x_s, s, e) + L_1 O(h^2)
\end{aligned} \qquad (19)$$

*where $(iii)$ hold according to the $\epsilon_\theta(x_{s \to t}, t) = \epsilon_\theta(x_s, s)$. When we set $e = 0$, the $G(x_s, s, 0)$ can be treated as diffusion model $D$ that directly predicts the original image $x_0$. We define $D(x_s, s) = G(x_s, s, 0)$ and $D(x_{s \to t}, t) = D(x_s, s) + L_1 O(h^2)$.*

*Let $e_n$ represent the error at $t_n$, which is defined as:*

$$e_n := D(x_{t_n}, t_n) - x_0. \qquad (20)$$

*We can derive the error at $t_{n+1}$ given the error at $t_n$:*

$$
\begin{aligned}
e_n =& D(x_{t_n}, t_n) - x_0 \\
=& D(x_{t_n}, t_n) - D(x_{t_{n-1} \to t_n}, t_n) + D(x_{t_{n-1} \to t_n}, t_n) - x_0 \\
=& \frac{x_{t_n} - \sigma_{t_n} \epsilon_\theta(x_{t_n}, t_n)}{\alpha_{t_n}} - \frac{x_{t_{n-1} \to t_n} - \sigma_{t_n} \epsilon_\theta(x_{t_{n-1} \to t_n}, t_n)}{\alpha_{t_n}} \\
& + D(x_{t_{n-1}}, t_{n-1}) - x_0 + L_1 O((t_n - t_{n-1})^2) \\
=& \frac{1}{\alpha_{t_n}}(x_{t_n} - x_{t_{n-1} \to t_n}) + \frac{\sigma_{t_n}}{\alpha_{t_n}}(\epsilon_\theta(x_{t_{n-1} \to t_n}, t_n) - \epsilon_\theta(x_{t_n}, t_n)) + e_{n-1} + L_1 O((t_n - t_{n-1})^2)
\end{aligned}
\tag{21}
$$

*According to the Lipschitz condition, we can further derive:*

$$
\begin{aligned}
||e_n|| \leq& \frac{1}{\alpha_{t_n}}||x_{t_n} - x_{t_{n-1} \to t_n}|| + \frac{\sigma_{t_n}}{\alpha_{t_n}}||\epsilon_\theta(x_{t_{n-1} \to t_n}, t_n) - \epsilon_\theta(x_{t_n}, t_n)|| + ||e_{n-1}|| + L_1 O((t_n - t_{n-1})^2) \\
\overset{(iv)}{\leq}& \frac{1}{\alpha_{t_n}} O((t_n - t_{n-1})^2) + \frac{\sigma_{t_n}}{\alpha_{t_n}} L ||x_{t_{n-1} \to t_n} - x_{t_n}|| + ||e_{n-1}|| + L_1 O((t_n - t_{n-1})^2) \\
\overset{(v)}{\leq}& ||e_{n-1}|| + K O((t_n - t_{n-1})^2)
\end{aligned}
\tag{22}
$$

*where $(iv), (v)$ hold according to the local error of Euler solver and Lipschitz condition of $\epsilon_\theta$. Therefore, we can derive the error recursively:*

$$
\begin{aligned}
||e_T|| \leq& K \sum_{i=1}^{N-1} O((t_i - t_{i-1})^2) \\
\leq& K \sum_{i=1}^{N-1} (t_i - t_{i-1}) O(\triangle) \\
\leq& K O(\triangle)(T - \xi)
\end{aligned}
\tag{23}
$$

*This shows that CTS is capable of achieving the same accuracy as multi-step approaches in a single-step generative frame work, thus demonstrating its efficiency and broad applicability for optimization-based generation. The proof is completed.*

**Proof 2:**

**Proof 3** *According to theorem 1 and CM (Song et al., 2023), the CTS loss is equivalent to consistency loss and the consistency function satisfies the Lipschitz condition. Given $L_{CTS}(\xi) = 0$, we have $L_{CD}(\xi) \leq O((\triangle t)^2)$ and $f_\theta(x_{s \to t}, t) \leq f_\theta(x_s, s) + O((\triangle t)^2)$. And $T \geq t_n \geq t_{n-1} \geq 0$.*

$$
\begin{aligned}
e_n =& f_\theta(x_{t_n}, t_n) - x_0 \\
=& f_\theta(x_{t_n}, t_n) - f_\theta(x_{t_{n-1} \to x_n}, t_n) + f_\theta(x_{t_{n-1} \to x_n}, t_n) - x_0 \\
\leq& f_\theta(x_{t_n}, t_n) - f_\theta(x_{t_{n-1} \to x_n}, t_n) + f_\theta(x_{t_{n-1}}, t_{n-1}) - x_0 + O((\triangle t)^2)
\end{aligned}
\tag{24}
$$

*According to the Lipschitz condition, we can further derive:*

$$
\begin{aligned}
||e_n|| \leq& ||f_\theta(x_{t_n}, t_n) - f_\theta(x_{t_{n-1} \to x_n}, x_n)|| + ||e_{n-1}|| + O((\triangle t)^2) \\
\overset{(vi)}{\leq}& L||x_{t_n} - x_{t_{n-1} \to x_n}|| + ||e_{n-1}|| + O((\triangle t)^2) \\
\overset{(vii)}{\leq}& L(O(t_n - t_{n+1})^2) + ||e_{n-1}|| + O((\triangle t)^2) \\
\leq& (L+1)O((t_n - t_{n+1})^2) + ||e_{n-1}|| \\
=& ||e_{n-1}|| + O((t_n - t_{n+1})^2)
\end{aligned}
\tag{25}
$$

where $(vi)$ and $(vii)$ hold according to the Lipschitz condition and local error of Euler solver respectively. Therefor, we can drive the error recursively:

$$
\begin{aligned}
||e_T|| &\leq \sum_{i=1}^{N-1} O((t_i - t_{i-1})^2) \\
&\leq \sum_{i=1}^{N-1} (t_i - t_{i-1}) O((\triangle)) \\
&\leq O((\triangle))(T - \xi)
\end{aligned}
\tag{26}
$$

The proof is completed.

**Corollary 1** *Based on Theorem. 1, we can derive two conclusions:*

- *Existing methods (Zhong et al., 2024; Chen et al., 2024b) which incorporate Latent Consistency Model for 3D generation can be amalgamated into our framework.*

- *Consistency function (Song et al., 2023) can ensures both self-consistency and cross-consistency.*

*Consistent3D (Wu et al., 2024) and CCD (Li et al., 2024b) drive the inspiration from Consistency model (Song et al., 2023), however, they are still trained based on Stable Diffusion-2-1 (Ho et al., 2020). Hence, we merely consider DreamLCM (Zhong et al., 2024) and Vividdreamer (Chen et al., 2024b) that based on LCM (Luo et al., 2023).*

**Proof 4** *DreamLCM (Zhong et al., 2024):*

$$
L_{DreamLCM} = \mathbb{E}_{t,c}[w(t)(\epsilon_\theta(x_t, t) - \epsilon)\frac{\partial g(\theta, c)}{\partial \theta}]
\tag{27}
$$

*where $x_t$ is obtained by Euler Solver from $x_s$ and $x_s = \alpha_s x_\pi + \sigma_s \epsilon$, $\epsilon \in N(0, I)$.*

*Vividreamer (Chen et al., 2024b):*

$$
L_{vividreamer} = \mathbb{E}_{t,c}[||w(t)(x_\pi - f(x_s, s))||_2^2]
\tag{28}
$$

*where $x_s = \alpha_s x_\pi + \sigma_s \epsilon$.*

*For DreamLCM (Zhong et al., 2024), according to eq. 15, $L_{CD}$ can express as:*

$$
\begin{aligned}
L_{CD} =& ||f_\theta(x_{t_n}, t_n) - f_\theta(x_{t_n \to t_{n+1}}, t_{n+1})||_2^2 \\
=& ||m + c_{skip}(t_{n+1}) O((\triangle t)^{p+1}) \\
& + c_{out}(t_n)\frac{\sigma_{t_{n+1}}}{\alpha_{t_{n+1}}}(\epsilon_\theta(x_{t_n \to t_{n+1}}, t_{n+1}) - \epsilon_\theta(x_{t_n}, t_n)) - \frac{O(h^2)}{\alpha_{t_{n+1}}} \\
& + m + [c_{out}(t_n) - c_{out}(t_{n+1})](\frac{\sigma_{t_n}}{\alpha_{t_n}})(\epsilon - \epsilon_\theta(x_{t_n}, t_n)) \\
& + [c_{out}(t_n) - c_{out}(t_{n+1})](\frac{\sigma_{t_{n+1}}}{\alpha_{t_{n+1}}}(\epsilon_\theta(x_{t_n}, t_n) - \epsilon_\theta(x_{t_n \to t_{n+1}}, t_{n+1})))||_2^2 \\
=& ||m + c_{skip}(t_{n+1}) O((\triangle t)^{p+1}) \\
& + c_{out}(t_{n+1})\frac{\sigma_{t_{n+1}}}{\alpha_{t_{n+1}}}(\epsilon_\theta(x_{t_n \to t_{n+1}}, t_{n+1}) - \epsilon_\theta(x_{t_n}, t_n)) - \frac{O(h^2)}{\alpha_{t_{n+1}}} \\
& + m + [c_{out}(t_n) - c_{out}(t_{n+1})](\frac{\sigma_{t_n}}{\alpha_{t_n}})(\epsilon - \epsilon_\theta(x_{t_n \to t_{n+1}}, t_{n+1}) + \epsilon_\theta(x_{t_n \to t_{n+1}}, t_{n+1}) - \epsilon_\theta(x_{t_n}, t_n))||_2^2 \\
=& m + c_{skip}(t_{n+1}) O((\triangle t)^{p+1}) \\
& + [c_{out}(t_{n+1})\frac{\sigma_{t_{n+1}}}{\alpha_{t_{n+1}}} + [c_{out}(t_n) - c_{out}(t_{n+1})](\frac{\sigma_{t_n}}{\alpha_{t_n}})](\epsilon_\theta(x_{t_n \to t_{n+1}}, t_{n+1}) - \epsilon_\theta(x_{t_n}, t_n)) \\
& + [c_{out}(t_n) - c_{out}(t_{n+1})](\frac{\sigma_{t_n}}{\alpha_{t_n}})\underbrace{(\epsilon - \epsilon_\theta(x_{t_n \to t_{n+1}}, t_{n+1})}_{loss\ of\ DreamLCM}
\end{aligned}
\tag{29}
$$

*DreamLCM assumes that the denoising process of LCM follows a smooth PF-ODE trajectory with a small slope. This assumption allows us to omit the term of $(\epsilon_\theta(x_{t_n \to t_{n+1}, t_{n+1}}) - \epsilon_\theta(x_{t_n}, t_n))$, therefore, in this assumption, $L_{DreamLCM}$ is a special case of $L_{CTS}$.*

*For Vividdreamer (Chen et al., 2024b), we have:*

$$
\begin{aligned}
L_{vividdreamer} =& ||x_\pi - f(x_s, s)||_2^2 \\
=& ||x_\pi - (c_{in}(s)x_\pi + c_{skip}(s)\frac{x_s - \sigma_s \epsilon_\theta(x_s, s)}{\alpha_s})||_2^2 \\
=& ||(1 - c_{skip}(s))x_\pi - c_{in}x_s + c_{skip}(s)x_\pi - c_{skip}(s)\frac{x_s - \sigma_s \epsilon_\theta(x_s, s)}{\alpha_s}||_2^2 \\
=& ||(1 - c_{skip}(s))x_\pi - c_{in}(\alpha_s x_\pi + \sigma_s \epsilon) + c_{skip}(s)\frac{\alpha_s x_\pi - \alpha_s x_\pi - \sigma_s(\epsilon - \epsilon_\theta(x_s, s))}{\alpha_s}||_2^2 \\
=& ||(1 - c_{skip}(s))x_\pi - c_{in}(s)(\alpha_s x_\pi + \sigma_s \epsilon) + c_{skip}(s)\frac{\sigma_s}{\alpha_s}(\epsilon - \epsilon_\theta(x_s, s))||_2^2 \\
=& ||[1 - c_{skip}(s) - c_{in}(s)\alpha_s]x_\pi - c_{in}(s)\sigma_s \epsilon + c_{skip}(s)\frac{\sigma_s}{\alpha_s}(\epsilon - \epsilon_\theta(x_s, s))||_2^2|| \\
\overset{(viii)}{\approx}& c_{skip}(s)\frac{\sigma_s}{\alpha_s}(\epsilon - \epsilon_\theta(x_s, s))||_2^2
\end{aligned}
$$

$$(30)$$

*where $(viii)$ hold according to the $c_{skip}(s) + c_{in}(s) \approx 1$ and $\mathbb{E}[\epsilon] = 0$. Therefore, Vividdreamer is a special case of our CTS loss.*

*According to the definition of self-consistency and cross-consistency(SDS) in previous methods (Wu et al., 2024; Li et al., 2024b), we can observe that $L_{CTS}$ simultaneously ensures two types of consistency:*

$$
L_{CTS} = \underbrace{||c_{out}(t_{n+1})(\frac{\sigma_{t_{n+1}}}{\alpha_{t_{n+1}}})(\epsilon_\theta(x_{t_n \to t_{n+1}}, t_{n+1}) - \epsilon_\theta(x_{t_n}, t_n))||_2^2}_{\textbf{\textit{self-consistency term}}}
$$
$$
+ \underbrace{||[c_{out}(t_{n+1}) - c_{out}(t_n)](\frac{\sigma_{t_n}}{\alpha_{t_n}})(\epsilon_\theta(x_{t_n}, t_n) - \epsilon)||_2^2}_{\textbf{\textit{cross-consistency term(SDS loss)}}}
$$

$$(31)$$

*The proof is complete.*

