# OpenReview forum: "SceneLCM: Multi-Room Indoor Scene Generation with Latent Consistency Modeling"
_ICLR.cc/2026/Conference — Submitted to ICLR 2026_

### Official Review · Reviewer_Cbt3 · 2025-10-23

**Soundness:** 2
**Presentation:** 1
**Contribution:** 2
**Rating:** 2
**Confidence:** 3

**Summary:**

This paper proposes a text-driven multi-room realistic indoor scene generation model. Specifically, a latent consistency model is used for layout generation, where a consistency trajectory sampling is designed to enable faster convergence of training and high-fidelity generation. Scene objects are generated in the form of 3D GS representation. Experimental results show high-quality and diverse indoor scene results.

**Strengths:**

+ The first to address multi-room generation
+ theoretical analysis of the proposed consistency trajectory ssampling

**Weaknesses:**

- The presentation of paper is poor. There are quite a lot assets that contributes to the model and it is not clear which is the key module that contributes the results.
- The motivation of the proposed consistency trajectory sampling is not clear explained. In L187-188, it is not clear about the Jacobian term and the mentioned auxiliary techniques such as perpendicular negative sampling.
- I do not find the justification of with and w/o the use of CTS loss in the visual format. It is hard to tell whether the term contributes or not solely from mathematical deductions.
- Missing citations of indoor layout generation papers. Is SubSection 3.3 one of the contribution of the paper?
- It is not clear how object-based 3D GS methods are used with the environment generation.
- Training data is not explained.
- How is user study is conducted?
- The main results of the proposed method shown in Figure 5 are not good compared to SceneTex, for example. Artifacts are shown here and there.
- Minor issues such as the apperance of L211.

**Questions:**

See above

---

> ### Author Response · Authors · 2025-11-15
>
> We sincerely apologize for delaying our response by two days, and we would like to thank you for your comments.
>
> 1. The key components are CTS loss and texture field, which we have emphasized in the introduction and abstract. We have dedicated separate chapters to elaborating on this part.
>
> 2. Ignoring the Jacobian term, perp-neg is a common technique in this field. It has been applied in almost all previous methods(luciddreamer[1], SDS[2], CSD[3], CFD[4], DreamLCM[5]). The Jacobian term refers to the item that ignores the derivative of the diffusion model with respect to the latent, which accelerates model generation, which is proposed in SDS[2]. Perp-neg is designed to address the multi-view inconsistency problem, and this method has been applied to all the aforementioned approaches.
>
> 3. I don't understand what "with CTS" and "without CTS" mean. If I don't use CTS, how can I generate objects and environment? This is not a plug-and-play method like dropout.
>
> 4. Due to space constraints, we have indeed omitted a lot of content, which will be added back in subsequent versions. SubSection 3.3  is our contribution, but not the main one.
>
> 5. We use Gaussians to represent furniture and meshes to represent scenes, respectively. After render mesh and gaussian, we blend them via depth map.
>
> 6. I don't understand that. We don't need to train the model, so we don't have the training dataset.
>
> 7. Find several strangers online, send the videos to them, and ask them to score. If you doubt the results of the user study, you can refer to several other automated evaluation metrics, which are all algorithm-based assessments. For example, CLIP and VQAscore.
>
> 8. SceneTex is completely different from ours in both task and experimental setup.
> * (1) Our task is completely different from SceneTex. SceneTex generates materials for given scene and object geometries, while we need to generate both materials and geometries of objects and scenes simultaneously. Such a comparison is unfair.
> * (2) our texture are obviously better than those of SceneTex. For example, in the Bohemian style, both SceneTex’s Figure 1 and ours have textures on the floor, but when you zoom in, you will find that our textures are much clearer.
> * (3) our method sis faster than sceneTex, we only take 1.2h for generation, while SceneTex requires 4 hours.
> * (4) SceneTex requires users to provide camera trajectories, and the 3D-FRONT dataset used by SceneTex includes camera trajectories. In contrast, we need to propose a camera trajectory that covers scenes of different scales.
> * (5) SceneTex cannot generate windows and curtains on meshes with only planes. In contrast, we can not only generate textures but also create wall decorations such as windows and curtains, making the room more complete. For details, you can refer to all images of SceneTex—none of their environments include generated windows or curtains.
> * (6) L211 is the title. I haven’t found any issues with it—could you please point them out in more detail?
>
> I’m not sure if my response has addressed your doubts, as these questions seem quite odd to me. After reading them, I’m unclear about what your main concern is and why you gave a score of 2. If you still have any unclear points, please raise them and discuss with me.

---

> > ### Author Response · Authors · 2025-11-15
> > **citation**
> >
> > [1] Liang, Yixun, et al. "Luciddreamer: Towards high-fidelity text-to-3d generation via interval score matching." Proceedings of the IEEE/CVF conference on computer vision and pattern recognition. 2024.
> >
> > [2] Poole, Ben, et al. "Dreamfusion: Text-to-3d using 2d diffusion." arXiv preprint arXiv:2209.14988 (2022).
> >
> > [3] Yu, Xin, et al. "Text-to-3d with classifier score distillation." ICLR 2024.
> >
> > [4] Yan, Runjie, Yinbo Chen, and Xiaolong Wang. "Consistent Flow Distillation for Text-to-3D Generation." The Thirteenth International Conference on Learning Representations.
> >
> > [5] Zhong, Yiming, et al. "Dreamlcm: Towards high quality text-to-3d generation via latent consistency model." Proceedings of the 32nd ACM International Conference on Multimedia. 2024.

---

> > ### Comment · Reviewer_Cbt3 · 2025-11-26
> >
> > Hi, thank you for your response, which have addressed my concerns. I will raise my score.

---

### Official Review · Reviewer_eXGD · 2025-11-01

**Soundness:** 3
**Presentation:** 3
**Contribution:** 2
**Rating:** 4
**Confidence:** 4

**Summary:**

The paper proposes SceneLCM, which combines LLM-driven layout generation with Latent Consistency Model (LCM)–based 3D optimization for automatic generation and interactive editing of single- and multi-room indoor scenes. The core technique is a new Consistency Trajectory Sampling (CTS) loss that maintains trajectory self-consistency during LCM distillation/sampling, accelerating convergence and improving fidelity; two theoretical results are provided (self-consistency and a bounded distillation error). The overall pipeline has three stages: (i) an LLM generates a floor plan and object configuration, which are then refined by iterative, executable program checks and cluster-based orientation assignment to resolve conflicts; (ii) objects are represented as 3D Gaussians and optimized with LCM+CTS; (iii) the environment uses a normal-aware multi-resolution texture field, optimized with LCM+CTS along a Zigzag camera trajectory for consistent texturing, while supporting texture/physically feasible edits and multi-room extension. Experiments demonstrate advantages in visual quality, multi-view consistency, and efficiency, with ablations verifying the key role of CTS.

**Strengths:**

1. Highly integrated, from end-to-end automation to interactivity: the pipeline unifies layout → objects → environment. In the layout stage, “programmatic verification + cluster-based orientation” reduces overlaps/empty space and mitigates orientation ambiguity, making the system practically robust.
2. Theory-backed CTS loss: within the consistency-model framework, CTS is derived from LCD, decomposes noise terms to remain compatible with techniques like Perp-Neg, and is supported by analyses of equivalence to the consistency objective and convergence properties.
3. Multi-room capability and coverage strategy: the proposed Zigzag trajectory adapts to different room scales, balancing surface normal coverage and overall coverage to stabilize texture optimization and improve consistency.
4. Normal-aware texture field: inspired by SceneTex and augmented with normal correlation, cross-attention transfers style between UV embeddings with similar normals, reducing tiling artifacts and multi-view inconsistency.

**Weaknesses:**

1.  How long does it take to generate a room, and how many tokens are consumed? The paper should explicitly report these figures.
2.  The examples suggest a relatively small number of objects per room. Can the authors evaluate scenes with larger object counts and report performance/quality when scaling up to denser rooms?
3. I’m particularly concerned about how this generative approach compares with other methods—for example, Holodeck’s object retrieval. Since generated materials are generally less accurate than assets retrieved directly from a database, what are the concrete advantages here? How does the overall speed compare? Please include examples for a side-by-side comparison.

**Questions:**

Please see the weaknesses.

---

> ### Author Response · Authors · 2025-11-15
>
> We sincerely apologize for delaying our response by two days, and we would like to thank you for your constructive comments.
> ### Runtime
> We will maintain an object repository to store all generated furniture. If there are furniture items in the room with the same style, similar size, and identical type, we will directly use such furniture instead of generating new ones.
>
> Thus, for a scene containing $N$ rooms and $M$ pieces of furniture, the maximum generation time is $N * 1.2h + M * 25mins / 3$(generate three pieces of furniture in parallel), the minimum generation time is $N * 1.h$ (all furniture can retrieval from reposity). For example, livingroom in Layout003 in our anonymous link https://scene-lcm.vercel.app/(provided in Supplementary Material), only the environment is generated for this room, and all furniture is obtained from the repository.
>
> ### larger scene
> 1. In fact, the furniture in our scenes is already more abundant than that in existing methods(holodeck[1], anyhome[2]), and our layout generation only adds Iterative Programmatic Verification and Cluster-Based Orientation Assignment on the basis of the existing method (AnyHome[2]).
>
> 2. Please check our anonymous link https://scene-lcm.vercel.app/, we have counted the number of furniture pieces for each layout. And in Figure.9, we compare with sota. And in supplment D.1, we use FFR to calculate the furniture proportion, indicating that the number of furniture in our scenes is higher.
>
> 3. We set the walls too high, which block some objects and make the scene appear sparse. For denser rooms, please check the computer room(22 objects) and dining room(12 objects) in Layout002, conference room(19 objects) in Layout003 in our anonymous link https://scene-lcm.vercel.app/.
>
> 4. **We must emphasize that there is a significant difference between floorplan-conditioned layout generation and non-floorplan-conditioned layout generation.** Other non-floorplan-conditioned methods such as Chat2Layout only need to handle the relationships between different furniture pieces. They can move overlapping objects freely. However, for floorplan-conditioned generation, LLMs need to simultaneously consider whether (the objects) exceed the boundaries. This is very difficult for LLMs because they usually lack spatial perception and distance awareness.
>
> 5. **If you think the room density in the room cases we provided above is insufficient, please let us know. We can conduct more experiments to prove the performance of our model.**
>
> ### compares with other methods
> #### Advantage
> * Environment:
>     * The object retrieval method cannot generate texture that are not available in the repository. For example, Van Gogh-style materials usually do not appear in the repository. In contrast, we could generate such style, please refer to Figure.15 in Page19.
>     * The textures generated by object retrieval are usually relatively simple and appear repeatedly, failing to achieve rich and refined textures like ours. Please refer to the texture of the study room in Figure 1 in regular paper: Chinese-style scenes contain striped textures on wall, while Bohemian-style scenes include diamond-shaped textures on the floor—these cannot be generated by the object retrieval method. In contrast, Infinigen indoors[3] or holodeck[1] only contain simple or Repeated texture, please refer to the Figure.1 in Infinigen indoors paper.
>
> * Object:
>     * Our CTS loss can generate not only furniture but also other non-furniture objects, please refer to Figure. 18 in Page 21 or sec 4 in anonymous link https://scene-lcm.vercel.app/. Additionally, furniture with specific styles is usually not included in the repository.
>     * If all objects in the scene need to be generated by LLMs and then queried in the repository, this is a huge challenge for LLMs—such as windows, curtains, carpets, etc. Currently, no method can achieve this. For example, Holodeck cannot generate curtains based on windows, nor can it generate carpets. In contrast, our curtains, windows, and carpets can be generated through CTS.
>     * In addition, the same object may have different scales in scenes of different styles. For example, windows only occupy a small part of the wall in modern-style scenes, while in Chinese or Japanese-style scenes, windows take up a larger area. However, LLMs usually only consider common sense and do not integrate style judgment when generating layouts (or there is no such method yet), resulting in deviations. Please refer to the chinese style room in Figure.1 in Page 1, chinese-style windows are usually rectangular and occupy 60-70% of the wall in height, while regular windows take up at most 50% of the wall height.
> #### Speed
> There is no doubt that the object retrieval method is faster than ours.

---

> > ### Comment · Reviewer_eXGD · 2025-11-27
> >
> > Thanks for the author's reply. I really appreciate that the authors provide a page to show results and answer my questions. Most concerns have been solved. I will raise my score.

---

> ### Author Response · Authors · 2025-11-15
> **citation**
>
> [1] Yang, Yue, et al. "Holodeck: Language guided generation of 3d embodied ai environments." Proceedings of the IEEE/CVF Conference on Computer Vision and Pattern Recognition. 2024.
>
> [2] Fu, Rao, et al. "Anyhome: Open-vocabulary generation of structured and textured 3d homes." European Conference on Computer Vision. Cham: Springer Nature Switzerland, 2024.
>
> [3] Raistrick, Alexander, et al. "Infinigen indoors: Photorealistic indoor scenes using procedural generation." Proceedings of the IEEE/CVF Conference on Computer Vision and Pattern Recognition. 2024.

---

### Official Review · Reviewer_Nqo4 · 2025-11-01

**Soundness:** 3
**Presentation:** 3
**Contribution:** 2
**Rating:** 4
**Confidence:** 4

**Summary:**

SceneLCM: Multi-Room Indoor Scene Generation with Latent Consistency Modeling proposes an automatic and interactive pipeline for generating complex, realistic, and physically coherent multi-room indoor scenes directly from textual descriptions. The framework unifies LLM-guided layout generation with Latent Consistency Model (LCM) optimization, addressing scalability, physical plausibility, and texture realism.

**Strengths:**

The proposed Consistency Trajectory Sampling (CTS) loss is both theoretically grounded and practically effective, offering a new formulation of consistency learning with provable self-consistency and bounded error. The approach is technically sound, with clear formulations, empirical results, and ablations.

**Weaknesses:**

While the proposed Consistency Trajectory Sampling (CTS) is an interesting and general contribution, the rest of the pipeline feels largely incremental relative to existing works in text-to-3D and layout-based scene generation. Beyond CTS, the integration of LLM-based layout generation, Gaussian-based object representation, and texture optimization largely combines known components from prior works rather than introducing fundamentally new mechanisms. The paper would benefit from explicitly articulating what distinguishes its pipeline components from existing work.

The Layout Generation stage, in particular, depends on repeated iterative regeneration by an LLM, which is computationally expensive and not conceptually novel. Many recent LLM-driven layout papers employ similar “generate–verify–regenerate” loops, and comparable strategies exist in traditional procedural layout synthesis. The paper should clarify whether SceneLCM introduces any principled advancement—e.g., improved convergence, better realism, or more accurate statistical matching to real-world layout distributions. Currently, it remains unclear how the generated layouts align with the true distribution of complex indoor environments, where the number, category, and spatial arrangement of objects are far richer than what prompt engineering can specify.

Parts of the pipeline also appear overly manual or rule-driven, such as the orientation assignment and verification steps. The method seems similar to existing rule-based procedural systems (e.g. [1]), except that the initialization is performed by an LLM; it is not evident how this hybrid approach achieves measurable gains in realism or diversity.

Figure 9 is labeled as a quantitative comparison but only shows one qualitative example per method, making it prone to cherry-picking and does not substantiate statistical improvement.

While CTS is a promising and theoretically grounded concept, the paper’s structure is not centered around CTS. The narrative currently treats it as a supporting component rather than the core contribution. A more focused presentation—where CTS is the centerpiece, and multi-room scene generation is framed as an application—would improve conceptual clarity. Additional ablations against more consistency or distillation baselines (e.g., SDS, ISM, CDS) across different tasks would also better demonstrate the generality of CTS beyond this specific application.

Section 3.3 is overly minimalistic, providing only high-level descriptions without sufficient methodological details.

[1] Raistrick, Alexander, et al. "Infinigen indoors: Photorealistic indoor scenes using procedural generation." Proceedings of the IEEE/CVF Conference on Computer Vision and Pattern Recognition. 2024.

**Questions:**

Could you provide details of Iterative Programmatic Verification and Cluster-Based Orientation Assignment?

Does CTS generalize to other tasks?

---

> ### Author Response · Authors · 2025-11-15
>
> We sincerely apologize for delaying our response by two days, and we would like to thank you for your constructive comments. We would like to respond to your feedback paragraph by paragraph as follows:
> ### Paragraph 1:  Distinguishes its pipeline components from existing work
> Text-to-3D methods for large-scale scenes can be divided into three categories (excluding object retrieval approaches): directly training a large model, directly utilizing existing generative models(SceneCraft[1], Luciddreamer[2], VideoScene[3]), or decomposing into a combination of distinct modules(DreamScene[4], GALA3D[5]). Our work follows the second line and incorporates layout generation for the sake of completeness.
>
> The first line of work cannot undergo subsequent processes such as editing or physical simulation, and they have the issue of multi-view consistency while the challenges of the second line of work lies in the integration of different modules, specifically in three aspects:
> * A loss function with higher scalability and efficiency that can simultaneously optimize scenes and objects.
>     * Please check Fig. 8 in regular paper, our loss function can stably optimize scenes and objects simultaneously while balancing speed and performance.
> * A camera trajectory with stronger scalability for layouts of different scales.
>     * The previous methods did not have this issue, as they could carefully design the layout or manually provide the camera trajectory.
> However, when layout generation is introduced, this issue will arise because the layout is no longer controllable at this point.  For example, large room: dining room of layout 001 in Figure. 19 in Page 22, small room: bedroom of layout002 in Figure. 20 in Page 22,  narrow room: conference room of Layout 003 in Figure. 21 in Page 23. Our camera trajectory can work on rooms of different scale.
> * How to generate complex textures, windows, curtains and other decorations in a plane-only environment while maintaining the integrity of the indoor space.
>     * Texture quality: Different styles correspond to different characteristic shapes. For examples, Japanese-style textures are composed of rectangles, Chinese-style textures feature stripes and window lattices, most Bohemian-style textures are dominated by diamonds, and Gypsy-style textures primarily use circles. This is a capability previous models lacked (e.g., DreamScene rendered all styles overly white with indistinct, similar textures).
>     * Our walls are just flat surfaces without complex geometry, yet our method can still generate windows, curtains, and other elements on these simply shaped planes. This is a feature that other methods do not possess. SceneCraft cannot generate windows or curtains, and their walls even lack textures; DreamScene can generate windows and curtains, but it has the issue of multi-view consistency. Please check Figure. 5 page 7, dreamscene fuses the bed and floor.
> * Additionally, our method effectively addresses the above three challenges, while also supporting editing and physical simulation. DreamScene can perform object editing but environment, and also they could not perform physical simulation(Their environment contains tens of millions of points, and direct simulation using PhyGaussians[6] requires enormous computational overhead, making it infeasible in implementation); SceneCraft cannot even segment objects.Please refer to Section 2: Environment Editing in our anonymous link (https://scene-lcm.vercel.app/). The corresponding anonymous link is contained in Anonymous URL.txt.
>
> ### Paragraph 2:  Layout Generation
> Layout generation is not the main contribution of our work; its contribution is very limited, which is why we only devoted a small portion of the text to it. This part is included primarily to make the entire task fully automatic.
>
> The main reason we included this part of the work is that existing multi-room generation schemes have many issues, and directly using them would only make things worse.
> * AnyHome[7] directly discards conflicting furniture, resulting in large areas of blank space. Therefore we introduce Iterative Programmatic Verification to keep these furniture.
> * Holodeck[8] uses a solver to adjust conflicting furniture, but its adjustments only satisfy preset rules without considering semantics. Therefore, it usually tends to place furniture along the walls, as this makes it easier to meet the rules. Therefore we propose Cluster-Based Orientation Assignment to preserve the semantics between furniture pieces.
> * About our improvement of layout, please check Figure. 9 in page 16, Figure. 10 in page 17, and Table. 2 in page 20, we use the metrics of Chat2Layout[9] for evaluation.
>
> Not enough space, please check the next answer.

---

> ### Author Response · Authors · 2025-11-15
> **Continuing from the previous response.**
>
> ### Paragraph 3:  Pipeline are manual or rule-driven and evident of realism or diversity
> We apologize for the misunderstanding caused by our writing. Our method is fully automatic with no manual intervention required, and layout generation is no exception. And our work is completely different from Infinigen Indoors[10].
> * Infinigen Indoors[10] is procedural and using only mathematical rules to generate. Three differences exist:
>     * (1) different task, Infinigen inputs a constraint program and we input text;
>     * (2) different method, Infinigen solve the constraint by solver, and we leverage LLM;
>     * (3) different capacity, infinigen can't generate objects or complex textures ​​which are not in procedural generator, and our generation has no restrictions. For example, in the images shown by Infinigen, the walls and floors are monochromatic and textureless. In contrast, our textures are more complex—the wall texture in the study room and the floor pattern in the entrance shown in Figure 1 of paper.
> * realistic: Layout-based object retrieval methods only consider layout generation and object querying; they do not take environment optimization into account and cannot generate complex texture like ours. For example, Infinigen Indoors[10] has almost no textures on its walls. For text-to-3D methods, we have already demonstrated the visual results and metrics. By the way,our environment optimization can be seamlessly integrated into layout-based methods.
> * diversity: I believe the diversity of our method **far exceeds** that of DreamScene and SceneCraft.
>     * SceneCraft is only fine-tuned on 80,000 images, and its generalization ability is almost non-existent. Please check Figure.5 and Figure. 6 in Page 7 and Page 8.
>     * DreamScene suffers from unwanted accumulated brightness in 3DGS[11], resulting in an almost uniform style across all its output styles. Please check our anonymous links: https://scene-lcm.vercel.app/, In sec. 7, we choose two difficult style: Industrial, baroque, only our method can achieve satisfactory results.
>     * Additionally, our model can also well distinguish styles with subtle differences. For example, Bohemian is usually composed of diamonds, while Gypsy is typically made up of circles. This is reflected in Bedroom Style 2 and Dining Room Style 1 under the Scene Editing section (Section 2) in the anonymous link: https://scene-lcm.vercel.app/, respectively. Japanese-style textures are composed of rectangles, Chinese-style textures feature stripes and window lattices, this is reflected in Section.6 in the anonymous link: https://scene-lcm.vercel.app/.
>
> ### Paragraph 4: cherry-picking example of Fig. 9
> * Due to limited space, we can only conduct visual comparison in this way. Please check our anonymous links: https://scene-lcm.vercel.app/, In sec. 7, we choose two difficult style: Industrial, baroqu, in this section, we provide a whole video for each methods.
> * substantiate statistical improvement: In fact, we have provided such metrics. Please refer to Table.1 in Page 9. We believe that for each scene, placing the camera at the center of the room, capturing 180 photos in a 360-degree circle, and conducting average estimation are sufficiently fair and comprehensive for evaluating performance. Moreover, except for the user study, all other evaluation metrics are automatically assessed, so there is no possibility of cherry-picking.
> * For each scene, we provide a video to show our visual results, please check our anonymous link: https://scene-lcm.vercel.app/.
> ### Paragraph 5: Suggestions on the paper structure
> Thank you very much for your suggestions. We will organize and revise the structure and content of the paper in response to your advice.

---

> ### Author Response · Authors · 2025-11-15
> **Continuing from the previous response.**
>
> ### Paragraph 6 \& Question 1
> We are very sorry that our writing failed to help you clearly understand this part. Since layout generation is not the core contribution, we have condensed the space.  First we generate a initial layout via LLM, then we apply Iterative Programmatic Verification and Orientation Assignment.
>
> Iterative Programmatic Verification
> * At first, we compute the furniture’s footprint. Proceeding to the next step is conditional on this footprint being ≤35% of the room area (a ratio consistent with real-world furniture distribution in room generation). If not, we direct the LLM to reduce furniture sizes.
> * Then we place the furniture into the room, and select furniture that overlaps with other objects or exceeds the layout boundaries. If the area of overlap is minimal, we’ll perform minor furniture adjustments. Otherwise, the selected furniture proceeds to the iterative programmatic verification.
> * We fix the size of the selected furniture and ask the LLM to adjust its position. If overlap or OOB persists after 5 iterations, we will unlock furniture size and instruct the LLM to adjust both size and position concurrently until the number of iterations exceeds 20 or no overlap and out-of-bounds(OOB) occur. In fact, we have never encountered a situation where adjustments were needed more than 5 times.
> * If the number of iterations exceeds 20, we regenerate the full layout. Therefore, we have only one boundary condition: when there is no overlap and no OOB occurs.
>
> Please check Figure. 9 in Page 16, anyhome contain much more furniture after Iterative Programmatic Verification. Noting that we only adjust a small number of objects each time, our success rate is higher and speed is faster than Holodeck’s.
>
> Orientation Assignment
> * Orientation correction simply involves resetting the orientation. For example, it determines the direction based on their orientation relative to the anchor. This process only requires one iteration.
>
> ### Question 2: Generalization of CTS
> CTS can be extended to related tasks such as object material generation, indoor scene material generation, and indoor scene editing.
>
> [1] Yang, Xiuyu, et al. "SceneCraft: Layout-guided 3D scene generation." Advances in Neural Information Processing Systems 37 (2024): 82060-82084.
>
> [2] Chung, Jaeyoung, et al. "Luciddreamer: Domain-free generation of 3d gaussian splatting scenes." arXiv preprint arXiv:2311.13384 (2023).
>
> [3] Wang, Hanyang, et al. "Videoscene: Distilling video diffusion model to generate 3d scenes in one step." 2025 IEEE/CVF Conference on Computer Vision and Pattern Recognition (CVPR). IEEE, 2025.
>
> [4] Li, Haoran, et al. "Dreamscene: 3d gaussian-based text-to-3d scene generation via formation pattern sampling." European Conference on Computer Vision. Cham: Springer Nature Switzerland, 2024.
>
> [5] Zhou, X., et al. "Gala3d: Towards text-to-3d complex scene generation via layout-guided generative gaussian splatting. arXiv 2024." arXiv preprint arXiv:2402.07207.
>
> [6] Xie, Tianyi, et al. "Physgaussian: Physics-integrated 3d gaussians for generative dynamics." Proceedings of the IEEE/CVF Conference on Computer Vision and Pattern Recognition. 2024.
>
> [7] Fu, Rao, et al. "Anyhome: Open-vocabulary generation of structured and textured 3d homes." European Conference on Computer Vision. Cham: Springer Nature Switzerland, 2024.
>
> [8] Yang, Yue, et al. "Holodeck: Language guided generation of 3d embodied ai environments." Proceedings of the IEEE/CVF Conference on Computer Vision and Pattern Recognition. 2024.
>
> [9] Wang, Can, et al. "Chat2Layout: Interactive 3D furniture layout with a multimodal LLM." arXiv preprint arXiv:2407.21333 (2024).
>
> [10] Raistrick, Alexander, et al. "Infinigen indoors: Photorealistic indoor scenes using procedural generation." Proceedings of the IEEE/CVF Conference on Computer Vision and Pattern Recognition. 2024.
>
> [11] Li, Zongrui, et al. "Connecting consistency distillation to score distillation for text-to-3d generation." European Conference on Computer Vision. Cham: Springer Nature Switzerland, 2024.

---

### Meta-Review · Area_Chair_qQtH · 2025-12-28

**Summary:**

The submission proposes SceneLCM, an automatic pipeline for single-/multi-room indoor scene generation from text by combining (i) LLM-based layout generation with programmatic verification and orientation assignment, (ii) object generation via 3D Gaussians optimized by an LCM, and (iii) environment texturing via a normal-aware texture field optimized along a Zigzag camera trajectory. Reviewers generally agree that the Consistency Trajectory Sampling (CTS) loss is the most technically substantive component, with useful theoretical grounding and strong qualitative results. However, multiple reviewers raised concerns that beyond CTS the pipeline largely integrates existing components (LLM “generate–verify–regenerate” layout, Gaussian-based optimization, SceneTex-like texture field ideas, trajectory heuristics), making the overall contribution appear incremental relative to concurrent text-to-3D and layout-based scene generation systems.

The rebuttal clarifies several missing implementation details (notably the iterative verification and orientation assignment) and addresses some practical questions (runtime discussion, denser-room examples, and motivation for comparisons). Still, the key issues remain: (1) novelty and conceptual clarity of the full system beyond CTS; (2) evaluation limitations, especially that quantitative comparisons are mainly for single-room settings with small prompt/scene counts and multi-room evidence is largely qualitative; and (3) insufficiently rigorous reporting of end-to-end cost (LLM token usage, full pipeline runtime across multi-room settings) and more systematic comparisons to retrieval-based multi-room systems. Given these remaining concerns, I lean towards reject.

**Reviewer Concerns:**

Concerns largely addressed in the rebuttal:
- Method details / missing clarity: The authors provided more concrete descriptions of Iterative Programmatic Verification and Cluster-Based Orientation Assignment, which were under-specified in the original draft.
- Practical questions (runtime / scaling / density): The rebuttal discusses runtime decomposition and a repository-based reuse strategy, and points to denser-room examples, addressing the concern that examples may be sparse.
- Clarification on “training data” and user study: The authors clarified the optimization-based nature (no training of a new generative model) and gave a brief description of the user study procedure.

Concerns that remain outstanding:
- Incrementality / novelty beyond CTS: While CTS appears promising, the overall pipeline still reads as an integration of known modules with several heuristic/rule-driven steps. The rebuttal argues for practicality and completeness, but does not fully establish a clear algorithmic or conceptual leap beyond prior multi-stage text-to-3D / indoor scene systems.
- Evaluation strength and scope: Quantitative evaluation is primarily single-room (limited prompts/rooms), and multi-room results are mostly qualitative. Even acknowledging baseline limitations, the paper would benefit from stronger multi-room evaluation (or alternative quantitative proxies) and broader evidence that CTS improves performance across tasks/settings, not only within this particular pipeline.
- Comparisons to retrieval-based multi-room systems: The rebuttal explains advantages (novel textures, style controllability, decorations) but the paper still lacks systematic side-by-side comparisons (quality + speed + controllability) against strong retrieval-based approaches in realistic multi-room regimes.
- Cost reporting and reproducibility: Token usage and end-to-end compute cost (including layout generation) are not clearly and consistently reported in the main paper. The user study description remains relatively light (participant recruitment, protocol, statistics).
- Presentation / positioning: Multiple reviewers noted that the narrative does not sufficiently center CTS and that the paper structure obscures what is truly novel. The rebuttal promises re-organization, but in the current version the presentation issues remain.

**Reviewer Scores:**

Reviewer **Nqo4**: likely stays at 4. The rebuttal answers the request for more detail on verification/orientation and argues for novelty, but the main concerns about incremental integration and evaluation strength remain, so I do not expect a decisive score increase.

Reviewer **eXGD**: indicated they would raise their score after the rebuttal; estimate 4 → 6, as runtime/scaling/comparison questions were addressed to their satisfaction.

Reviewer **Cbt3**: also indicated they would raise their score; estimate 2 → 4, since several confusions about the pipeline and comparisons were clarified, though the paper’s presentation and evaluation limitations likely keep it below a confident accept.

---

### Decision · Program_Chairs · 2026-01-26

Reject